# MICN: Multi-scale Local and Global context modeling for Long-term Series Forecasting

**Huiqiang Wang**[1]**, Jian Peng**[1]**, Feihu Huang**[1✉]**, Jince Wang**[1]**, Junhui Chen**[1]**, Yifei Xiao**[2]

[1]College of Computer Science, Sichuan University, China
[2]School of Computer Science and Engineering,
University of Electronic Science and Technology of China, China
{wanghuiqiang,wjc,chenjunhui}@stu.scu.edu.cn,
{jianpeng,huangfh}@scu.edu.cn, xyf1989@uestc.edu.cn

## Abstract

Recently, Transformer-based methods have achieved surprising performance in the field of long-term series forecasting, but the attention mechanism for computing global correlations entails high complexity. And they do not allow for targeted modeling of local features as CNN structures do. To solve the above problems, we propose to combine local features and global correlations to capture the overall view of time series (e.g., fluctuations, trends). To fully exploit the underlying information in the time series, a multi-scale branch structure is adopted to model different potential patterns separately. Each pattern is extracted with down-sampled convolution and isometric convolution for local features and global correlations, respectively. In addition to being more effective, our proposed method, termed as Multi-scale Isometric Convolution Network (**MICN**), is more efficient with linear complexity about the sequence length with suitable convolution kernels. Our experiments on six benchmark datasets show that compared with state-of-the-art methods, MICN yields 17.2% and 21.6% relative improvements for multivariate and univariate time series, respectively. Code is available at `https://github.com/wanghq21/MICN`.

## 1 Introduction

Researches related to time series forecasting are widely applied in the real world, such as sensor network monitoring (Papadimitriou & Yu., 2006), weather forecasting, economics and finance (Zhu & Shasha, 2002), and disease propagation analysis (Matsubara et al., 2014) and electricity forecasting. In particular, long-term time series forecasting is increasingly in demand in reality. Therefore, this paper focuses on the task of long-term forecasting. The problem to be solved is to predict values for a future period: $X_{t+1}, X_{t+2}, ..., X_{t+T-1}, X_{t+T}$ , based on observations from a historical period: $X_1, X_2, ..., X_{t-1}, X_t$, and $T \gg t$.

As a classic CNN-based model, TCN (Bai et al., 2018) uses causal convolution to model the temporal causality and dilated convolution to expand the receptive field. It can integrate the local information of the sequence better and achieve competitive results in short and medium-term forecasting (Sen et al., 2019) (Borovykh et al., 2017). However, limited by the receptive field size, TCN often needs many layers to model the global relationship of time series, which greatly increases the complexity of the network and the training difficulty of the model.

Transformers (Vaswani et al., 2017) based on the attention mechanism shows great power in sequential data, such as natural language processing (Devlin et al., 2019) (Brown et al., 2020), audio processing (Huang et al., 2019) and even computer vision (Dosovitskiy et al., 2021) (Liu et al., 2021b). It has also recently been applied in long-term series forecasting tasks (Li et al., 2019b) (Wen et al., 2022) and can model the long-term dependence of sequences effectively, allowing leaps and bounds in the accuracy and length of time series forecasts (Zhu & Soricut, 2021) (Wu et al., 2021b) (Zhou et al., 2022). The learned attention matrix represents the correlations between different time points of the sequence and can explain relatively well how the model makes future predictions based on past information. However, it has a quadratic complexity, and many of the computations

between token pairs are non-essential, so it is also an interesting research direction to reduce its computational complexity. Some notable models include: LogTrans (Li et al., 2019b), Informer (Zhou et al., 2021), Reformer (Kitaev et al., 2020), Autoformer Wu et al. (2021b), Pyraformer (Liu et al., 2021a), FEDformer (Zhou et al., 2022).

However, as a special sequence, time series has not led to a unified modeling direction so far. In this paper, we combine the modeling perspective of CNNs with that of Transformers to build models from the realistic features of the sequences themselves, i.e., local features and global correlations. Local features represent the characteristics of a sequence over a small period $T$, and global correlations are the correlations exhibited between many periods $T_1, T_2, ...T_{n-1}, T_n$. For example, the temperature at a moment is not only influenced by the specific change during the day but may also be correlated with the overall trend of a period (e.g., week, month, etc.). We can identify the value of a time point more accurately by learning the overall characteristics of that period and the correlation among many periods before. Therefore, a good forecasting method should have the following two properties: (1) The ability to extract local features to measure short-term changes. (2) The ability to model the global correlations to measure the long-term trend.

Based on this, we propose Multi-scale Isometric Convolution Network (**MICN**). We use multiple branches of different convolution kernels to model different potential pattern information of the sequence separately. For each branch, we extract the local features of the sequence using a local module based on downsampling convolution, and on top of this, we model the global correlation using a global module based on isometric convolution. Finally, Merge operation is adopted to fuse information about different patterns from several branches. This design reduces the time and space complexity to linearity, eliminating many unnecessary and redundant calculations. MICN achieves state-of-the-art accuracy on five real-world benchmarks. The contributions are summarized as follows:

- We propose MICN based on convolution structure to efficiently replace the self-attention, and it achieves linear computational complexity and memory cost.

- We propose a multiple branches framework to deeply mine the intricate temporal patterns of time series, which validates the need and validity for separate modeling when the input data is complex and variable.

- We propose a local-global structure to implement information aggregation and long-term dependency modeling for time series, which outperforms the self-attention family and Auto-correlation mechanism. We adopt downsampling one-dimensional convolution for local features extraction and isometric convolution for global correlations discovery.

- Our empirical studies show that the proposed model improves the performance of state-of-the-art method by 17.2% and 21.6% for multivariate and univariate forecasting, respectively.

## 2 RELATED WORK

### 2.1 CNNS AND TRANSFORMERS

Convolutional neural networks (CNN) are widely used in computer vision, natural language processing and speech recognition (Sainath et al., 2013) (Li et al., 2019a) (Han et al., 2020). It is widely believed that this success is due to the use of convolution operations, which can introduce certain inductive biases, such as translation invariance, etc. CNN-based methods are usually modeled from the local perspective, and convolution kernels can be very good at extracting local information from the input. By continuously stacking convolution layers, the field of perception can be extended to the entire input space, enabling the aggregation of the overall information.

Transformer (Vaswani et al., 2017) has achieved the best performance in many fields since its emergence, which is mainly due to the attention mechanism. Unlike modeling local information directly from the input, the attention mechanism does not require stacking many layers to extract global information. Although the complexity is higher and learning is more difficult, it is more capable of learning long-term dependencies (Vaswani et al., 2017).

Although CNNs and Transformers are modeled from different perspectives, they both aim to achieve efficient utilization of the overall information of the input. In this paper, from the view of combining

the modeling principles of CNNs and Transformers, we consider both local and global context, extract local features of data first, and then model global correlation on this basis. Furthermore, our method achieves lower computational effort and complexity.

## 2.2 MODELING BOTH LOCAL AND GLOBAL CONTEXT

Both local and global relationships play an important role in sequence modeling. Some works have been conducted to study how to combine local and global modeling into a unified model to achieve high efficiency and interpretability. Two well-known architectures are: Conformer (Gulati et al., 2020) and Lite Transformer (Wu et al., 2020).

Conformer is a variant of Transformer and has achieved state-of-the-art performance in many speech applications. It adopts the attention mechanism to learn the global interaction, the convolution module to capture the relative-offset-based local features, and combines these two modules sequentially. However, Conformer does not analyze in detail what local and global features are learned and how they affect the final output. There is also no explanation why the attention module is followed by a convolution module. Another limitation of Conformer is the quadratic complexity with respect to the sequence length due to self-attention.

Lite Transformer also adopts convolution to extract local information and self-attention to capture long-term correlation, but it separates them into two branches for parallel processing. A visual analysis of the feature weights extracted from the two branches is also presented in the paper, which can provide a good interpretation of the model results. However, the parallel structure of the two branches determines that there may be some redundancy in its computation, and it still has the limitation of quadratic complexity.

Whether the convolution and self-attention are serialized to extract local and global relationships step by step or in parallel, it inevitably results in quadratic time and space complexity. Therefore, in this paper, we propose a new framework for modeling local features and global correlations of time series along with a new module instead of attention mechanism. We also use the convolution operation to extract its local information and then propose isometric convolution to model the global correlation between each segment of the local features. This modeling method not only avoids more redundant computations but also reduces the overall time and space complexity to linearity with respect to the sequence length.

## 3 MODEL

### 3.1 MICN FRAMEWORK

The overall structure of MICN is shown in Figure 1. The long time series prediction task is to predict a future series of length $O$ based on a past series of length $I$, which can be expressed as $input - I - predict - O$, where $O$ is much larger than $I$. Inspired by traditional time series decomposition algorithms (Robert et al., 1990) (Wu et al., 2021b), we design a multi-scale hybrid decomposition (**MHDecomp**) block to separate complex patterns of input series. Then we use **Seasonal Prediction Block** to predict seasonal information and **Trend-cyclical Prediction Block** to predict trend-cyclical information. Then add the prediction results up to get the final prediction $Y_{pred}$. We donate $d$ as the number of variables in multivariate time series and $D$ as the hidden state of the series. The details will be given in the following sections.

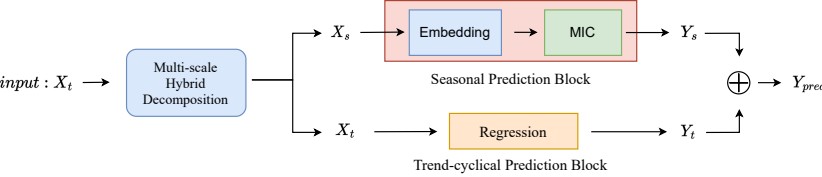

Figure 1: MICN overall architecture.

## 3.2 MULTI-SCALE HYBRID DECOMPOSITION

Previous series decomposition algorithms (Wu et al., 2021b) adopt the moving average to smooth out periodic fluctuations and highlight the long-term trends. For the input series $X \in R^{I \times d}$, the process is:

$$
\begin{aligned}
X_t &= AvgPool(Padding(X))_{kernel} \\
X_s &= X - X_t,
\end{aligned}
\tag{1}
$$

where: $X_t, X_s \in R^{I \times d}$ denote the trend-cyclical and seasonal parts, respectively. The use of the $Avgpool(\cdot)$ with the padding operation keeps the series length unchanged. But the parameter *kernel* of the $Avgpool(\cdot)$ is artificially set and there are often large differences in trend-cyclical series and seasonal series obtained from different *kernels*. Therefore, we design a multi-scale hybrid decomposition block that uses several different *kernels* of the $Avgpool(\cdot)$ and can separate several different patterns of trend-cyclical and seasonal parts purposefully. Different from the MOEDecomp block of FEDformer (Zhou et al., 2022), we use simple mean operation to integrate these different patterns because we cannot determine the weight of each pattern before learning its features. Correspondingly, we put this weighting operation in the Merge part of Seasonal Prediction block after the representation of the features. Concretely, for the input series $X \in R^{I \times d}$, the process is:

$$
\begin{aligned}
X_t &= mean(AvgPool(Padding(X))_{kernel_1}, ..., AvgPool(Padding(X))_{kernel_n}) \\
X_s &= X - X_t,
\end{aligned}
\tag{2}
$$

where $X_t, X_s \in R^{I \times d}$ denote the trend-cyclical and seasonal part, respectively. The different kernels are consistent with multi-scale information in Seasonal Prediction block. The effectiveness is demonstrated experimentally in Appendix B.1.

## 3.3 TREND-CYCLICAL PREDICTION BLOCK

Currently, Autoformer (Wu et al., 2021b) concatenates the mean of the original series and then accumulates it with the trend-cyclical part obtained from the inner series decomposition block. But there is no explanation of this and no proof of its effectiveness. In this paper, we use a simple linear regression strategy to make a prediction about trend-cyclical, demonstrating that simple modeling of trend-cyclical is also necessary for non-stationary series forecasting tasks (See Section 4.2). Concretely, for the trend-cyclical series $X_t \in R^{I \times d}$ obtained with MHDecomp block, the process is:

$$
Y_t^{regre} = regression(X_t)
\tag{3}
$$

where $Y_t^{regre} \in R^{O \times d}$ denotes the prediction of the trend part using the linear regression strategy. And we use MICN-regre to represent MICN model with this trend-cyclical prediction method.

For comparison, we use the mean of $X_t$ to cope with the series where the trend-cyclical keeps constant:

$$
Y_t^{mean} = mean(X_t)
\tag{4}
$$

where $Y_t^{mean} \in R^{O \times d}$ denotes the prediction of the trend part. And we use MICN-mean to represent MICN model with this trend-cyclical prediction method.

## 3.4 SEASONAL PREDICTION BLOCK

As shown in Figure 2, the Seasonal Prediction Block focuses on the more complex seasonal part modeling. After embedding the input sequence $X_s$, we adopt multi-scale isometric convolution to capture the local features and global correlations, and branches of different scales model different underlying patterns of the time series. We then merge the results from different branches to complete comprehensive information utilization of the sequence. It can be summarised as follows:

$$
\begin{aligned}
X_s^{emb} &= Embedding(Concat(X_s, X_{zero})) \\
Y_s^0 &= X_s^{emb} \\
Y_{s,l} &= MIC(Y_{s,l-1}), \quad l \in \{1, 2, ..., N\} \\
Y_s &= Truncate(Projection(Y_{s,N})),
\end{aligned}
\tag{5}
$$

where $X_{zero} \in R^{O \times d}$ denotes the placeholders filled with zero, and $X_s^{emb} \in R^{(I+O) \times D}$ denotes the embedded representation of $X_s$. $Y_{s,l} \in R^{(I+O) \times D}$ represents the output of $l-th$ multi-scale isometric

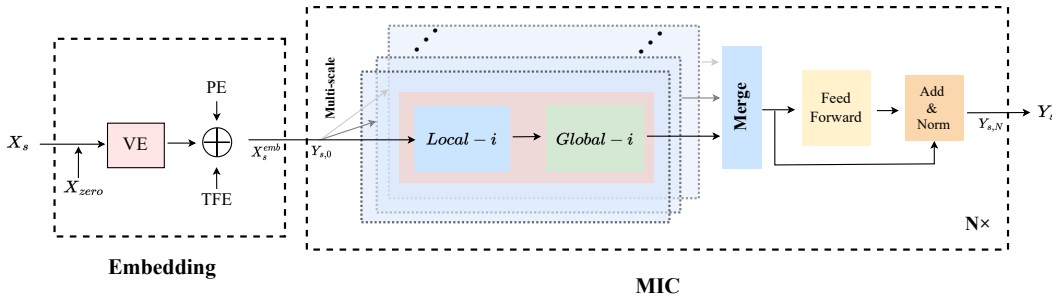

Figure 2: Seasonal Prediction Block.

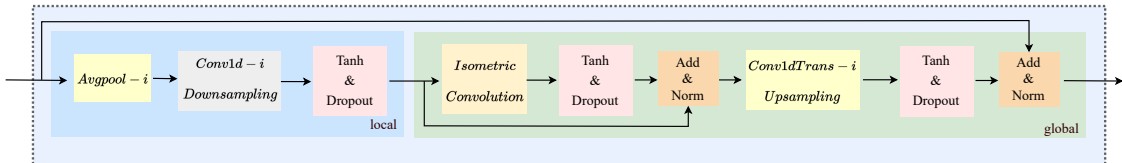

Figure 3: Local-Global module architecture.

convolution (MIC) layer, and $Y_s \in R^{O \times d}$ represents the final prediction of the seasonal part after a linear function Projection with $Y_{s,N} \in R^{(I+O) \times D}$ and Truncate operation. The detailed description of Embedding and MIC will be given as follows.

**Embedding**   The decoder of the latest Transformer-based models such as Informer (Zhou et al., 2021), Autoformer (Wu et al., 2021b) and FEDformer (Zhou et al., 2022) contain the latter half of the encoder's input with the length $\frac{I}{2}$ and placeholders with length O filled by scalars, which may lead to redundant calculations. To avoid this problem and adapt to the prediction length $O$, we replace the traditional encoder-decoder style input with a simpler complementary 0 strategy. Meanwhile, we follow the setting of FEDformer and adopt three parts to embed the input. The process is:

$$X_s^{emb} = sum(TFE + PE + VE(Concat(X_s, X_{zero}))) \tag{6}$$

where $X_s^{emb} \in R^{(I+O) \times D}$. $TFE$ represents time features encoding (e.g., MinuteOfHour, HourOfDay, DayOfWeek, DayOfMonth, and MonthOfYear), $PE$ represents positional encoding and $VE$ represents value embedding.

**Multi-scale isometric Convolution(MIC) Layer**   MIC layer contains several branches, with different scale sizes used to model potentially different temporal patterns. In each branch, as shown in Figure 3, the local-global module extracts the local features and the global correlations of the sequence (See Appendix B.7 for more detailed description). Concretely, after obtaining the corresponding single pattern by $avgpool$, the local module adopts one-dimensional convolution to implement downsampling. The process is:

$$
\begin{aligned}
Y_{s,l} &= Y_{s,l-1} \\
Y_{s,l}^{local,i} &= Conv1d(Avgpool(Padding(Y_{s,l}))_{kernel=i})_{kernel=i},
\end{aligned}
\tag{7}
$$

where $Y_{s,l-1}$ denotes the output of $(l-1)-th$ MIC layer and $Y_{s,0} = X_s^{emb}$. $i \in \left\{ \frac{I}{4}, \frac{I}{8}, ... \right\}$ denote the different scale sizes corresponding to the different branches in Figure 2. For Conv1d, we set $stride = kernel = i$, which serves as compression of local features. $Y_{s,l}^{local,i} \in R^{\frac{(I+O)}{i} \times D}$ represents the result obtained by compressing local features, which is a short sequence.

And furthermore, the global module is designed to model the global correlations of the output of the local module. A commonly used method for modeling global correlations is the self-attention mechanism. But in this paper, we use a variant of casual convolution, isometric convolution, as an alternative. As shown in Figure 4, isometric convolution pads the sequence of length $S$ with

placeholders zero of length $S-1$ , and its kernel is equal to $S$ . It means that we can use a large convolution kernel to measure the global correlation of the whole series. The current generative prediction approach is to add placeholder to the input sequence, which has no actual sequence information in the second half. The Isometric Convolution can enable sequential inference of sequences by fusing local features information. Moreover, the kernel of Isometric convolution is determined by all the training data, which can introduces a global temporal inductive bias (translation equivariance, etc.) and achieve better generalization than self-attention (the correlations are obtained from the product between different elements). Meanwhile, we demonstrate that for a shorter sequence, isometric convolution is superior to self-attention. The detailed experiments of the proof are in Appendix B.3. And to keep the sequence length constant, we upsample the result of the isometric convolution using transposed convolution. The global module can be formalized as follows:

$$Y_{s,l}^{',i} = Norm(Y_{s,l}^{local,i} + Dropout(Tanh(IsometricConv(Y_{s,l}^{local,i}))))$$
$$Y_{s,l}^{global,i} = Norm(Y_{s,l-1} + Dropout(Tanh(Conv1dTranspose(Y_{s,l}^{',i})_{kernel=i}))),$$
(8)

where $Y_{s,l}^{local,i} \in R^{\frac{(I+O)}{i} \times D}$ denote the result after the global correlations modeling. $Y_{s,l-1}$ is the output of $l-1$ MIC layer. $Y_{s,l}^{global,i} \in R^{(I+O) \times D}$ represents the result of this pattern (i.e., this branch).

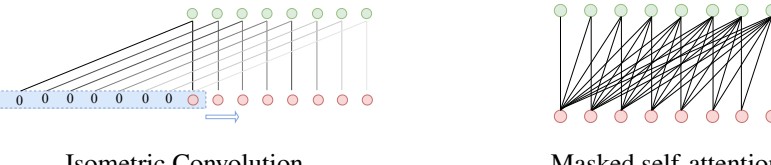

Isometric Convolution          Masked self-attention

Figure 4: Isometric Convolution architecture vs. Masked self-attention architecture

Then we propose to use Conv2d to merge the different patterns with different weights instead of the traditional Concat operation. The validity of this weighting approach is verified in Appendix B.4. The process is:

$$Y_{s,l}^{merge} = (Conv2d(Y_{s,l}^{global,i}, i \in \left\{\frac{I}{4}, \frac{I}{8}, ...\right\}))$$
$$Y_{s,l} = Norm(Y_{s,l}^{merge} + FeedForward(Y_{s,l}^{merge})),$$
(9)

where $Y_{s,l} \in R^{(I+O) \times D}$ represents the result of $l-th$ MIC layer.

To get the final prediction of the seasonal part, we use the projection and truncate operations:

$$Y_s = Truncate(Projection(Y_{s,N})) \tag{10}$$

where $Y_{s,N} \in R^{(I+O) \times D}$ represents the output of N-th MIC layer, and $Y_s \in R^{O \times d}$ represents the final prediction about the seasonal part.

## 4 EXPERIMENTS

**Dataset** To evaluate the proposed MICN, we conduct extensive experiments on six popular real-world datasets, covering many aspects of life: energy, traffic, economics, and weather. We follow standard protocol (Zhou et al., 2021) and split all datasets into training, validation and test set in chronological order by the ratio of 6:2:2 for the ETT dataset and 7:1:2 for the other datasets. More details about the datasets and implementation are described in Appendix A.1 and A.2.

**Baselines** We include four transformer-based models: FEDformer (Zhou et al., 2022), Autoformer (Wu et al., 2021b), Informer (Zhou et al., 2021), LogTrans (Li et al., 2019b), two RNN-based models: LSTM (Hochreiter & Schmidhuber, 1997), LSTNet (Lai et al., 2018b) and CNN-based model TCN (Bai et al., 2018) as baselines. For the univariate setting, we mainly compare transformer-based models. For the state-of-the-art model FEDformer, we compare the better one (FEDformer-f).

Table 1: **Multivariate** long-term series forecasting results with input length $I = 96$ and prediction length $O \in \{96, 192, 336, 720\}$ (for ILI, the input length $I = 36$). A lower MSE or MAE indicates a better prediction, and the best results are highlighted in bold.

| Methods | | MICN-regre | | MICN-mean | | FEDformer | | Autoformer | | Informer | | LogTrans | | LSTNet | | LSTM | | TCN | |
|---|---|---|---|---|---|---|---|---|---|---|---|---|---|---|---|---|---|---|---|
| Metric | | MSE | MAE | MSE | MAE | MSE | MAE | MSE | MAE | MSE | MAE | MSE | MAE | MSE | MAE | MSE | MAE | MSE | MAE |
| ETTm2 | 96 | **0.179** | **0.275** | 0.203 | 0.287 | 0.203 | 0.287 | 0.255 | 0.339 | 0.365 | 0.453 | 0.768 | 0.642 | 3.142 | 1.365 | 2.041 | 1.073 | 3.041 | 1.330 |
| | 192 | 0.307 | 0.376 | **0.262** | **0.326** | 0.269 | 0.328 | 0.281 | 0.340 | 0.533 | 0.563 | 0.989 | 0.757 | 3.154 | 1.369 | 2.249 | 1.112 | 3.072 | 1.339 |
| | 336 | 0.325 | 0.388 | **0.305** | **0.353** | 0.325 | 0.366 | 0.339 | 0.372 | 1.363 | 0.887 | 1.334 | 0.872 | 3.160 | 1.369 | 2.568 | 1.238 | 3.105 | 1.348 |
| | 720 | 0.502 | 0.490 | **0.389** | **0.407** | 0.421 | 0.415 | 0.422 | 0.419 | 3.379 | 1.388 | 3.048 | 1.328 | 3.171 | 1.368 | 2.720 | 1.287 | 3.135 | 1.354 |
| Electricity | 96 | **0.164** | **0.269** | 0.193 | 0.308 | 0.193 | 0.308 | 0.201 | 0.317 | 0.274 | 0.368 | 0.258 | 0.357 | 0.680 | 0.645 | 0.375 | 0.437 | 0.985 | 0.813 |
| | 192 | **0.177** | **0.285** | 0.200 | 0.308 | 0.201 | 0.315 | 0.222 | 0.334 | 0.296 | 0.386 | 0.266 | 0.368 | 0.725 | 0.676 | 0.442 | 0.473 | 0.996 | 0.821 |
| | 336 | **0.193** | **0.304** | 0.219 | 0.328 | 0.214 | 0.329 | 0.231 | 0.338 | 0.300 | 0.394 | 0.280 | 0.380 | 0.828 | 0.727 | 0.439 | 0.473 | 1.000 | 0.824 |
| | 720 | **0.212** | **0.321** | 0.224 | 0.332 | 0.246 | 0.355 | 0.254 | 0.361 | 0.373 | 0.439 | 0.283 | 0.376 | 0.957 | 0.811 | 0.980 | 0.814 | 1.438 | 0.784 |
| Exchange | 96 | **0.102** | **0.235** | 0.173 | 0.297 | 0.148 | 0.278 | 0.197 | 0.323 | 0.847 | 0.752 | 0.968 | 0.812 | 1.551 | 1.058 | 1.453 | 1.049 | 3.004 | 1.432 |
| | 192 | **0.172** | **0.316** | 0.324 | 0.408 | 0.271 | 0.380 | 0.300 | 0.369 | 1.204 | 0.895 | 1.040 | 0.851 | 1.477 | 1.028 | 1.846 | 1.179 | 3.048 | 1.444 |
| | 336 | **0.272** | **0.407** | 0.639 | 0.598 | 0.460 | 0.500 | 0.509 | 0.524 | 1.672 | 1.036 | 1.659 | 1.081 | 1.507 | 1.031 | 2.136 | 1.231 | 3.113 | 1.459 |
| | 720 | **0.714** | **0.658** | 1.218 | 0.862 | 1.195 | 0.841 | 1.447 | 0.941 | 2.478 | 1.310 | 1.941 | 1.127 | 2.285 | 1.243 | 2.984 | 1.427 | 3.150 | 1.458 |
| Traffic | 96 | **0.519** | **0.309** | 0.575 | 0.344 | 0.587 | 0.366 | 0.613 | 0.388 | 0.719 | 0.391 | 0.684 | 0.384 | 1.107 | 0.685 | 0.843 | 0.453 | 1.438 | 0.784 |
| | 192 | **0.537** | **0.315** | 0.580 | 0.349 | 0.604 | 0.373 | 0.616 | 0.382 | 0.696 | 0.379 | 0.685 | 0.390 | 1.157 | 0.706 | 0.847 | 0.453 | 1.463 | 0.794 |
| | 336 | **0.534** | **0.313** | 0.583 | 0.345 | 0.621 | 0.383 | 0.622 | 0.337 | 0.777 | 0.420 | 0.733 | 0.408 | 1.216 | 0.730 | 0.853 | 0.455 | 1.479 | 0.799 |
| | 720 | **0.577** | **0.325** | 0.601 | 0.363 | 0.626 | 0.382 | 0.660 | 0.408 | 0.864 | 0.472 | 0.717 | 0.396 | 1.481 | 0.805 | 1.500 | 0.805 | 1.499 | 0.804 |
| Weather | 96 | **0.161** | **0.229** | 0.183 | 0.250 | 0.217 | 0.296 | 0.266 | 0.336 | 0.300 | 0.384 | 0.458 | 0.490 | 0.594 | 0.587 | 0.369 | 0.406 | 0.615 | 0.589 |
| | 192 | **0.220** | **0.281** | 0.246 | 0.317 | 0.276 | 0.336 | 0.307 | 0.367 | 0.598 | 0.544 | 0.658 | 0.589 | 0.560 | 0.565 | 0.416 | 0.435 | 0.629 | 0.600 |
| | 336 | **0.278** | **0.331** | 0.293 | 0.335 | 0.339 | 0.380 | 0.359 | 0.395 | 0.578 | 0.523 | 0.797 | 0.652 | 0.597 | 0.587 | 0.455 | 0.454 | 0.639 | 0.608 |
| | 720 | **0.311** | **0.356** | 0.373 | 0.399 | 0.403 | 0.428 | 0.419 | 0.428 | 1.059 | 0.741 | 0.869 | 0.675 | 0.618 | 0.599 | 0.535 | 0.520 | 0.639 | 0.610 |
| ILI | 24 | **2.684** | **1.112** | 3.029 | 1.180 | 3.228 | 1.260 | 3.483 | 1.287 | 5.764 | 1.677 | 4.480 | 1.444 | 6.026 | 1.770 | 5.914 | 1.734 | 6.624 | 1.830 |
| | 36 | 2.667 | 1.068 | **2.507** | **1.013** | 2.679 | 1.080 | 3.103 | 1.148 | 4.755 | 1.467 | 4.799 | 1.467 | 5.340 | 1.668 | 6.631 | 1.845 | 6.858 | 1.879 |
| | 48 | 2.558 | 1.052 | **2.423** | **1.012** | 2.622 | 1.078 | 2.669 | 1.085 | 4.763 | 1.469 | 4.800 | 1.468 | 6.080 | 1.787 | 6.736 | 1.857 | 6.968 | 1.892 |
| | 60 | 2.747 | 1.110 | **2.653** | **1.085** | 2.857 | 1.157 | 2.770 | 1.125 | 5.264 | 1.564 | 5.278 | 1.560 | 5.548 | 1.720 | 6.870 | 1.879 | 7.127 | 1.918 |

Table 2: **Univariate** long-term series forecasting results with input length $I = 96$ and prediction length $O \in \{96, 192, 336, 720\}$ (for ILI, the input length $I = 36$). A lower MSE or MAE indicates a better prediction, and the best results are highlighted in bold.

| Methods | | MICN-regre | | MICN-mean | | FEDformer | | Autoformer | | Informer | | LogTrans | |
|---|---|---|---|---|---|---|---|---|---|---|---|---|---|
| Metric | | MSE | MAE | MSE | MAE | MSE | MAE | MSE | MAE | MSE | MAE | MSE | MAE |
| ETTm2 | 96 | **0.059** | **0.176** | 0.074 | 0.206 | 0.072 | 0.206 | 0.065 | 0.189 | 0.088 | 0.225 | 0.075 | 0.208 |
| | 192 | 0.100 | 0.234 | **0.098** | **0.238** | 0.102 | 0.245 | 0.118 | 0.256 | 0.132 | 0.283 | 0.129 | 0.275 |
| | 336 | 0.153 | 0.301 | **0.135** | **0.282** | 0.130 | 0.279 | 0.154 | 0.305 | 0.180 | 0.336 | 0.154 | 0.302 |
| | 720 | 0.210 | 0.354 | **0.175** | **0.326** | 0.178 | 0.325 | 0.182 | 0.335 | 0.300 | 0.435 | 0.160 | 0.321 |
| Electricity | 96 | 0.310 | 0.398 | 0.326 | 0.418 | **0.253** | **0.370** | 0.341 | 0.438 | 0.484 | 0.538 | 0.288 | 0.393 |
| | 192 | 0.300 | 0.394 | 0.317 | 0.410 | **0.282** | **0.386** | 0.345 | 0.428 | 0.557 | 0.558 | 0.432 | 0.483 |
| | 336 | **0.323** | **0.413** | 0.376 | 0.450 | 0.346 | 0.431 | 0.406 | 0.470 | 0.636 | 0.613 | 0.430 | 0.483 |
| | 720 | **0.364** | **0.449** | 0.417 | 0.479 | 0.422 | 0.484 | 0.565 | 0.581 | 0.819 | 0.682 | 0.491 | 0.531 |
| Exchange | 96 | **0.099** | **0.240** | 0.179 | 0.312 | 0.154 | 0.304 | 0.241 | 0.387 | 0.591 | 0.615 | 0.237 | 0.377 |
| | 192 | **0.198** | **0.354** | 0.304 | 0.420 | 0.286 | 0.420 | 0.300 | 0.369 | 1.183 | 0.912 | 0.738 | 0.619 |
| | 336 | **0.302** | **0.447** | 0.711 | 0.651 | 0.511 | 0.555 | 0.509 | 0.524 | 1.367 | 0.984 | 2.018 | 1.070 |
| | 720 | **0.738** | **0.662** | 1.416 | 0.918 | 1.301 | 0.879 | 1.260 | 0.867 | 1.872 | 1.072 | 2.405 | 1.175 |
| Traffic | 96 | **0.158** | **0.241** | 0.214 | 0.324 | 0.207 | 0.312 | 0.246 | 0.346 | 0.257 | 0.353 | 0.226 | 0.317 |
| | 192 | **0.154** | **0.236** | 0.228 | 0.336 | 0.205 | 0.312 | 0.266 | 0.370 | 0.299 | 0.376 | 0.314 | 0.408 |
| | 336 | **0.165** | **0.243** | 0.217 | 0.337 | 0.219 | 0.323 | 0.263 | 0.371 | 0.312 | 0.387 | 0.387 | 0.453 |
| | 720 | **0.182** | **0.264** | 0.225 | 0.339 | 0.244 | 0.344 | 0.269 | 0.372 | 0.366 | 0.436 | 0.491 | 0.437 |
| Weather | 96 | **0.0029** | **0.039** | 0.0038 | 0.052 | 0.0062 | 0.062 | 0.011 | 0.081 | 0.0038 | 0.044 | 0.0046 | 0.052 |
| | 192 | 0.0021 | 0.034 | **0.0015** | **0.029** | 0.0060 | 0.062 | 0.0075 | 0.067 | 0.0023 | 0.040 | 0.0056 | 0.060 |
| | 336 | **0.0023** | **0.034** | 0.0039 | 0.053 | 0.0041 | 0.050 | 0.0063 | 0.062 | 0.0041 | 0.049 | 0.0060 | 0.054 |
| | 720 | 0.0048 | 0.054 | **0.0024** | **0.037** | 0.0055 | 0.059 | 0.0085 | 0.070 | 0.0031 | 0.042 | 0.0071 | 0.063 |
| ILI | 24 | 0.674 | 0.671 | **0.607** | **0.587** | 0.708 | 0.627 | 0.948 | 0.732 | 5.282 | 2.050 | 3.607 | 1.662 |
| | 36 | 0.712 | 0.733 | **0.551** | **0.604** | 0.584 | 0.617 | 0.634 | 0.650 | 4.554 | 1.916 | 2.407 | 1.363 |
| | 48 | 0.823 | 0.803 | **0.693** | **0.704** | 0.717 | 0.697 | 0.791 | 0.752 | 4.273 | 1.846 | 3.106 | 1.575 |
| | 60 | 0.992 | 0.892 | **0.816** | **0.779** | 0.855 | 0.774 | 0.874 | 0.797 | 5.214 | 2.057 | 3.698 | 1.733 |

## 4.1 MAIN RESULTS

**Multivariate results** For multivariate long-term series forecasting, MICN achieves the state-of-the-art performance in all benchmarks and all prediction length settings (Table 1). Compared to the previous best model FEDformer, MICN yields an 17.2% averaged MSE reduction. Especially, under the input-96-predict-96 setting, MICN gives 12% relative MSE reduction in ETTm2, 14% relative MSE reduction in Electricity, 31% relative MSE reduction in Exchange, 12% relative MSE reduction in Traffic, 26% relative MSE reduction in Weather, 17% relative MSE reduction in ILI, and 18.6% average MSE reduction in this setting. We can also find that MICN makes consistent improvements as the prediction increases, showing its competitiveness in terms of long-term time-series forecasting. Note that MICN still provides remarkable improvements with a 51% averaged MSE reduction in the Exchange dataset that is without obvious periodicity. All above shows that MICN can cope well with a variety of time-series forecasting tasks in real-world applications. More results about other ETT benchmarks are provided in Appendix A.3. See Appendix C.3 for detailed showcases.

**Univariate results**    We also show the univariate time-series forecasting results in Table 2. Significantly, MICN achieves a 21.6% averaged MSE reduction compared to FEDformer. Especially for the Weather dataset, MICN gives 53% relative MSE reduction under the predict-96 setting, 75% relative MSE reduction under the predict-192 setting, 44% relative MSE reduction under the predict-336 setting, and 56% relative MSE reduction under the predict-720 setting. It again verifies the greater time-series forecasting capacity. More results about other ETT benchmarks are provided in Appendix A.3. See Appendix C.2 for detailed showcases.

## 4.2 ABLATION STUDIES

**Trend-cyclical Prediction Block**    We attempt to verify the necessity of modeling the trend-cyclical part when using a decomposition-based structure. Like Autoformer (Wu et al., 2021b), previous methods decompose the time series and then take the mean prediction of the trend information, which is then added to the other trend information obtained from the decomposition module in the model. However, the reasons and rationality are not argued in the relevant papers. In this paper, we use simple linear regression to predict the trend-cyclical part and we also record the results of the mean prediction for comparison. Note that with different trend-cyclical prediction blocks, we have different models named MICN-regre and MICN-mean. As shown in Table 3, MICN-regre performs better than MICN-mean overall. Because all the datasets are non-stationary, a simple modeling for trend-cyclical to give model a holistic view of the trend direction is necessary. See Appendix B.2 for more visualization results and analysis.

Table 3: Comparison of sample linear regression prediction and mean prediction in multivariate datasets. The better results are highlighted in bold.

| Datasets | | ETTm2 | | | | Electricity | | | | Exchange | | | | Traffic | | | | WTH | | | |
|---|---|---|---|---|---|---|---|---|---|---|---|---|---|---|---|---|---|---|---|---|---|
| Prediction Length O | | 96 | 192 | 336 | 720 | 96 | 192 | 336 | 720 | 96 | 192 | 336 | 720 | 96 | 192 | 336 | 720 | 96 | 192 | 336 | 720 |
| MICN - regre | MSE | **0.179** | 0.307 | 0.325 | 0.502 | **0.164** | **0.177** | **0.193** | **0.212** | **0.102** | **0.172** | **0.272** | **0.714** | **0.519** | **0.537** | **0.534** | **0.577** | **0.161** | **0.220** | **0.278** | **0.311** |
| | MAE | **0.275** | 0.376 | 0.388 | 0.490 | **0.269** | **0.285** | **0.304** | **0.321** | **0.235** | **0.316** | **0.407** | **0.658** | **0.309** | **0.315** | **0.313** | **0.325** | **0.229** | **0.281** | **0.331** | **0.356** |
| MICN - mean | MSE | 0.200 | **0.262** | **0.305** | **0.389** | 0.188 | 0.200 | 0.219 | 0.224 | 0.173 | 0.324 | 0.639 | 1.218 | 0.575 | 0.580 | 0.583 | 0.601 | 0.183 | 0.246 | 0.293 | 0.373 |
| | MAE | 0.287 | **0.326** | **0.353** | **0.407** | 0.302 | 0.308 | 0.328 | 0.332 | 0.297 | 0.408 | 0.598 | 0.862 | 0.344 | 0.349 | 0.345 | 0.363 | 0.250 | 0.317 | 0.335 | 0.399 |

**Local-Global Structure vs. Auto-correlation, self-attention**    In this work, we propose the local-global module to model the underlying pattern of time series, including local features and global correlations, while the previous outstanding model Autoformer uses auto-correlation. We replace the auto-correlation module in the original Autoformer with our proposed local-global module (we set $i \in \{12, 16\}$) for training, and the results are shown in Table 4. Also, We replace the Local-Global module in MICN-regre with the Auto-Correlation module and self-attention module for training, and the results are shown in Table 5. They all demonstrate that modeling time series in terms of local features and global correlations is better and more realistic.

Table 4: Ablation of Local-global structure in other models. We **replace the Auto-Correlation in Autoformer with our local-global module** and implement it in the multivariate Electricity, Exchange and Traffic. The better results are highlighted in bold.

| Datasets | | Electricity | | | | Exchange | | | | Traffic | | | |
|---|---|---|---|---|---|---|---|---|---|---|---|---|---|
| Prediction Length O | | 96 | 192 | 336 | 720 | 96 | 192 | 336 | 720 | 96 | 192 | 336 | 720 |
| Autoformer-Local-Global | MSE | **0.192** | **0.204** | **0.223** | **0.238** | 0.194 | **0.293** | 1.012 | 1.289 | **0.572** | **0.580** | **0.587** | **0.601** |
| | MAE | **0.314** | **0.323** | **0.339** | **0.352** | 0.338 | 0.416 | 0.766 | 0.928 | **0.352** | **0.351** | **0.353** | **0.359** |
| Autoformer Auto-correlation | MSE | 0.207 | 0.236 | 0.275 | 0.289 | **0.160** | 0.327 | **0.509** | **1.133** | 0.675 | 0.666 | 0.765 | 1.098 |
| | MAE | 0.323 | 0.343 | 0.372 | 0.380 | **0.292** | **0.415** | **0.527** | **0.825** | 0.406 | 0.425 | 0.487 | 0.647 |

## 4.3 MODEL ANALYSIS

**Impact of input length**    In time series forecasting tasks, the size of the input length indicates how much historical information the algorithm can utilize. In general, a model that has a strong ability to model long-term temporal dependency should perform better as the input length increases. Therefore, we conduct experiments with different input lengths and the same prediction length to validate our model. As shown in Figure 5, when the input length is relatively long, the performance of Transformer-based models becomes worse because of repeated short-term patterns as stated in (Zhou et al., 2021). Relatively, the overall performance of MICN prediction gradually gets better as

Table 5: Ablation of Local-global structure in our model. We **replace the Local-Global module in MICN-regre with Auto-correlation and self-attention** and implement it in the multivariate Electricity, Exchange and Traffic. The better results are highlighted in bold.

| Datasets | | Electricity | | | | Exchange | | | | Traffic | | | |
|---|---|---|---|---|---|---|---|---|---|---|---|---|---|
| Prediction Length $O$ | | 96 | 192 | 336 | 720 | 96 | 192 | 336 | 720 | 96 | 192 | 336 | 720 |
| MICN- | MSE | **0.164** | **0.177** | **0.193** | **0.212** | **0.102** | **0.172** | **0.272** | **0.714** | **0.519** | **0.537** | **0.534** | **0.577** |
| Local-Global | MAE | **0.269** | **0.285** | **0.304** | **0.321** | **0.235** | 0.316 | **0.407** | **0.658** | **0.309** | **0.315** | **0.313** | **0.325** |
| MICN- | MSE | 0.205 | 0.209 | 0.229 | 0.260 | 0.111 | 0.178 | 0.331 | 0.804 | 0.596 | 0.613 | 0.609 | 0.635 |
| Auto-Correlation | MAE | 0.299 | 0.305 | 0.327 | 0.353 | 0.255 | **0.311** | 0.440 | 0.718 | 0.366 | 0.386 | 0.379 | 0.381 |
| MICN | MSE | 0.181 | 0.194 | 0.216 | 0.271 | 0.147 | 0.290 | 0.480 | 1.578 | 0.612 | 0.642 | 0.622 | 0.656 |
| self-attention | MAE | 0.289 | 0.304 | 0.321 | 0.362 | 0.291 | 0.402 | 0.549 | 0.978 | 0.357 | 0.376 | 0.374 | 0.382 |

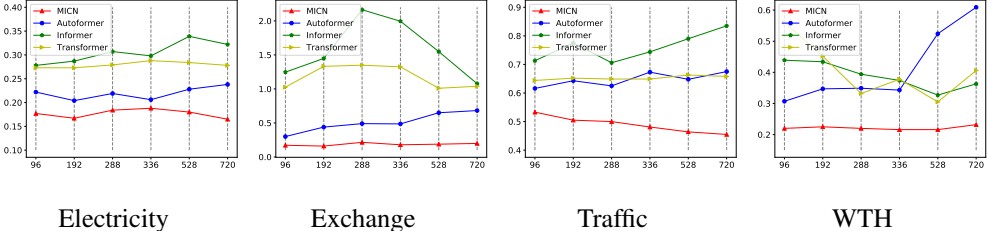

| Electricity | Exchange | Traffic | WTH |
|---|---|---|---|

Figure 5: The MSE results with different input lengths and same prediction lengths (192 time steps).

the input length increases, indicating that MICN can capture the long-term temporal dependencies well and extract useful information deeply.

**Robustness analysis**  We use a simple noise injection to demonstrate the robustness of our model. Concretely, we randomly select data with proportion $\varepsilon$ in the original input sequence and randomly perturb the selected data in the range $[-2X_i, 2X_i]$, where $X_i$ denotes the original data. The data after noise injection is then trained, and the MSE and MAE metrics are recorded. The results are shown in Table 6. As the proportion of perturbations $\varepsilon$ increases, the MSE and MAE metrics of the predictions increase by a small amount. It indicates that MICN exhibits good robustness in response to less noisy data (up to 10%) and has a great advantage in dealing with many data abnormal fluctuations (e.g. abnormal power data caused by equipment damage).

Table 6: Robustness analysis of **multivariate** results. Different $\varepsilon$ indicates different proportions of noise injection. And MICN-regre is used as the base model.

| Datasets | | Electricity | | | | Exchange | | | | Traffic | | | |
|---|---|---|---|---|---|---|---|---|---|---|---|---|---|
| Prediction Length $O$ | | 96 | 192 | 336 | 720 | 96 | 192 | 336 | 720 | 96 | 192 | 336 | 720 |
| MICN - | MSE | 0.164 | 0.177 | 0.193 | 0.212 | 0.102 | 0.172 | 0.272 | 0.714 | 0.519 | 0.537 | 0.534 | 0.577 |
| regre | MAE | 0.269 | 0.285 | 0.304 | 0.321 | 0.235 | 0.316 | 0.407 | 0.658 | 0.309 | 0.315 | 0.313 | 0.325 |
| $\varepsilon = 1\%$ | MSE | 0.163 | 0.179 | 0.192 | 0.217 | 0.103 | 0.172 | 0.289 | 0.691 | 0.518 | 0.530 | 0.535 | 0.575 |
| | MAE | 0.270 | 0.288 | 0.303 | 0.325 | 0.237 | 0.316 | 0.424 | 0.652 | 0.321 | 0.312 | 0.315 | 0.323 |
| $\varepsilon = 5\%$ | MSE | 0.164 | 0.181 | 0.192 | 0.218 | 0.104 | 0.167 | 0.296 | 1.742 | 0.518 | 0.541 | 0.558 | 0.585 |
| | MAE | 0.272 | 0.289 | 0.303 | 0.328 | 0.239 | 0.308 | 0.413 | 1.009 | 0.313 | 0.327 | 0.330 | 0.328 |
| $\varepsilon = 10\%$ | MSE | 0.171 | 0.189 | 0.202 | 0.220 | 0.136 | 0.181 | 0.402 | 0.944 | 0.538 | 0.557 | 0.561 | 0.605 |
| | MAE | 0.281 | 0.297 | 0.311 | 0.328 | 0.273 | 0.324 | 0.497 | 0.771 | 0.332 | 0.324 | 0.325 | 0.335 |

## 5  CONCLUSIONS

This paper presents a convolution-based framework MICN, which makes predictions for the trend-cyclical part and seasonal part separately. It achieves $\mathcal{O}(LD^2)$ complexity and yields consistent state-of-the-art performance in extensive real-world datasets. In the Seasonal-Prediction block, we use different scales to mine the sequence for potentially different patterns, each modeled from a local and global perspective, which is implemented by different convolution operations. The proposed isometric convolution outperforms self-attention in terms of capturing global correlations for a short sequence. The extensive experiments further demonstrate the effectiveness of our modeling approach for long-term forecasting tasks.

ACKNOWLEDGEMENTS

This work was supported by the Sichuan Science and Technology Program (2023YFG0112) and the National Key R&D Program of China (2020YFB0704502) and the Sichuan Science and Technology Program (2022YFG0034) and Postdoctoral Interdisciplinary Innovation Fund(10822041A2137) and Sichuan University and Yibin Cooperation Program (2020CDYB-30).

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

# A  SUPPLEMENTAL EXPERIMENTS

## A.1  DATASET DETAILS

In this work, the details of the experiment datasets are summarized as follows: (1) *ETT* (Zhou et al., 2021) dataset contains two visions of the sub-dataset: ETTh and ETTm, collected from electricity transformers every 15 minutes and 1 hour between July 2016 and July 2018. (2) *Electricity*[1] dataset contains the electricity consumption of 321 customers recorded hourly from 2012 to 2014. (3) *Exchange* (Lai et al., 2018a) dataset records daily exchange rates of eight different countries daily ranging from 1990 to 2016. (4) *Traffic*[2] contains the data from California Department of Transportation hourly, which describes the road occupancy rates measured by different sensors on San Francisco Bay area freeways. (5) *Weather*[3] contains 21 meteorological indicators, recorded every 10 minutes for 2020 whole year. (6) *ILI*[4] records influenza-like illness (ILI) patients data weekly from Centers for Disease Control and Prevention of the United States between 2002 and 2021. Table 7 summarizes feature details (Sequence Length: Len, Dimension: Dim, Frequency: Freq) .

Table 7: The details of datasets.

| Dataset | len | dim | freq |
|---|---|---|---|
| ETTh | 17420 | 8 | 1h |
| ETTm | 69680 | 8 | 15 min |
| Electricity | 26304 | 322 | 1h |
| Exchange | 7588 | 9 | 1 day |
| Traffic | 17544 | 863 | 1h |
| Weather | 52696 | 22 | 10 min |
| ILI | 966 | 8 | 7 day |

## A.2  IMPLEMENTATION DETAILS

Our method is trained with the L2 loss, using the ADAM optimizer with an initial learning rate of $10^{-3}$. Batch size is set to 32. The training process is early stopped after three epochs if there is no loss degradation on the valid set. The mean square error (MSE) and mean absolute error (MAE) are used as metrics. All the experiments are repeated 3 times with different seeds, implemented in PyTorch and conducted on NVIDIA RTX A5000 24GB GPU. The hyper-parameter $i$ is set to $\{12, 16\}$, and the hyper-parameter sensitivity analysis can be seen in Appendix A.4 . For a fairer comparison, we fix the input length to 96 for all datasets (36 for ILI). MICN contains 1 MIC layer. We use MICN-regre and MICN-mean to represent the different strategies of trend-cyclical prediction block in the following.

## A.3  FULL BENCHMARK ON THE ETT DATASETS

We build the benchmark on the four ETT datasets in Table 8 and Table 9. The ETTh1 and ETTh2 datasets are recorded hourly, while the ETTm1 and ETTm2 datasets are recorded every 15 minutes. MICN achieves state-of-the-art performance in all benchmarks in general. Especially for the multivariate ETTm1 dataset, MICN gives 17% relative MSE reduction under the predict-96 setting, gives 15% relative MSE reduction under the predict-192 setting, gives 8% relative MSE reduction under the predict-336 setting, gives 15% relative MSE reduction under the predict-720 setting.

## A.4  HYPER-PARAMETER SENSITIVITY

As shown in Table 10, we can verify the model robustness with respect to hyper-parameter $i$. Different values of $i$ have slightly different results. Concretely, when $i$ take one value, MICN performs worse

---

[1] https://archive.ics.uci.edu/ml/datasets/ElectricityLoadDiagrams20112014
[2] http://pems.dot.ca.gov
[3] https://www.bgc-jena.mpg.de/wetter/
[4] https://gis.cdc.gov/grasp/fluview/fluportaldashboard.html

Table 8: Multivariate long-term forecasting results on ETT full benchmark. The best results are highlighted in bold.

| Methods | | MICN-regre | | MICN-mean | | FEDformer | | Autoformer | | Informer | | LogTrans | |
|---|---|---|---|---|---|---|---|---|---|---|---|---|---|
| Metric | | MSE | MAE | MSE | MAE | MSE | MAE | MSE | MAE | MSE | MAE | MSE | MAE |
| ETTh1 | 96 | 0.421 | 0.431 | 0.398 | 0.427 | **0.376** | **0.419** | 0.449 | 0.459 | 0.865 | 0.713 | 0.878 | 0.740 |
| | 192 | 0.474 | 0.487 | 0.430 | 0.453 | **0.420** | **0.448** | 0.500 | 0.482 | 1.008 | 0.792 | 1.037 | 0.824 |
| | 336 | 0.569 | 0.551 | **0.440** | **0.460** | 0.459 | 0.465 | 0.521 | 0.496 | 1.107 | 0.809 | 1.238 | 0.932 |
| | 720 | 0.770 | 0.672 | **0.491** | **0.509** | 0.506 | 0.507 | 0.514 | 0.512 | 1.181 | 0.865 | 1.135 | 0.852 |
| ETTh2 | 96 | **0.299** | **0.364** | 0.332 | 0.377 | 0.346 | 0.388 | 0.358 | 0.397 | 3.755 | 1.525 | 2.116 | 1.197 |
| | 192 | 0.441 | 0.454 | **0.422** | **0.441** | 0.429 | 0.439 | 0.456 | 0.452 | 5.602 | 1.931 | 4.315 | 1.635 |
| | 336 | 0.654 | 0.567 | **0.447** | **0.474** | 0.496 | 0.487 | 0.482 | 0.486 | 4.721 | 1.835 | 1.124 | 1.604 |
| | 720 | 0.956 | 0.716 | **0.442** | **0.467** | 0.463 | 0.474 | 0.515 | 0.511 | 3.647 | 1.625 | 3.188 | 1.540 |
| ETTm1 | 96 | **0.316** | **0.362** | 0.360 | 0.399 | 0.379 | 0.419 | 0.505 | 0.475 | 0.672 | 0.571 | 0.600 | 0.546 |
| | 192 | **0.363** | **0.390** | 0.402 | 0.426 | 0.426 | 0.441 | 0.553 | 0.496 | 0.795 | 0.669 | 0.837 | 0.700 |
| | 336 | **0.408** | **0.426** | 0.403 | 0.437 | 0.445 | 0.459 | 0.621 | 0.537 | 1.212 | 0.871 | 1.124 | 0.832 |
| | 720 | 0.481 | 0.476 | **0.459** | **0.464** | 0.543 | 0.490 | 0.671 | 0.561 | 1.166 | 0.823 | 1.153 | 0.820 |
| ETTm2 | 96 | **0.179** | **0.275** | 0.203 | 0.287 | 0.203 | 0.287 | 0.255 | 0.339 | 0.365 | 0.453 | 0.768 | 0.642 |
| | 192 | 0.307 | 0.376 | **0.262** | **0.326** | 0.269 | 0.328 | 0.281 | 0.340 | 0.533 | 0.563 | 0.989 | 0.757 |
| | 336 | 0.325 | 0.388 | **0.305** | **0.353** | 0.325 | 0.366 | 0.339 | 0.372 | 1.363 | 0.887 | 1.334 | 0.872 |
| | 720 | 0.502 | 0.490 | **0.389** | **0.407** | 0.421 | 0.415 | 0.422 | 0.419 | 3.379 | 1.338 | 3.048 | 1.328 |

Table 9: Univariate long-term forecasting results on ETT full benchmark. The best results are highlighted in bold.

| Methods | | MICN-regre | | MICN-mean | | FEDformer | | Autoformer | | Informer | | LogTrans | |
|---|---|---|---|---|---|---|---|---|---|---|---|---|---|
| Metric | | MSE | MAE | MSE | MAE | MSE | MAE | MSE | MAE | MSE | MAE | MSE | MAE |
| ETTh1 | 96 | **0.058** | **0.186** | 0.069 | 0.210 | 0.079 | 0.215 | 0.071 | 0.206 | 0.193 | 0.377 | 0.283 | 0.468 |
| | 192 | **0.079** | **0.210** | 0.081 | 0.223 | 0.104 | 0.245 | 0.114 | 0.262 | 0.217 | 0.395 | 0.234 | 0.409 |
| | 336 | **0.092** | **0.237** | 0.104 | 0.259 | 0.119 | 0.270 | 0.107 | 0.258 | 0.202 | 0.381 | 0.386 | 0.546 |
| | 720 | 0.138 | 0.298 | **0.090** | **0.238** | 0.142 | 0.299 | 0.126 | 0.283 | 0.183 | 0.355 | 0.475 | 0.628 |
| ETTh2 | 96 | 0.155 | 0.300 | 0.137 | 0.286 | **0.128** | **0.271** | 0.153 | 0.306 | 0.213 | 0.373 | 0.217 | 0.379 |
| | 192 | **0.169** | **0.316** | 0.179 | 0.334 | 0.185 | 0.330 | 0.204 | 0.351 | 0.227 | 0.387 | 0.281 | 0.429 |
| | 336 | 0.238 | 0.384 | **0.203** | **0.359** | 0.231 | 0.378 | 0.246 | 0.389 | 0.242 | 0.401 | 0.293 | 0.437 |
| | 720 | 0.447 | 0.561 | **0.193** | **0.352** | 0.278 | 0.420 | 0.268 | 0.409 | 0.291 | 0.439 | 0.218 | 0.387 |
| ETTm1 | 96 | **0.033** | **0.134** | 0.039 | 0.152 | 0.033 | 0.140 | 0.056 | 0.183 | 0.109 | 0.277 | 0.049 | 0.171 |
| | 192 | **0.048** | **0.164** | 0.050 | 0.180 | 0.058 | 0.186 | 0.081 | 0.216 | 0.151 | 0.310 | 0.157 | 0.317 |
| | 336 | 0.079 | 0.210 | **0.064** | **0.202** | 0.084 | 0.231 | 0.076 | 0.218 | 0.427 | 0.591 | 0.289 | 0.459 |
| | 720 | 0.096 | 0.233 | **0.085** | **0.232** | 0.102 | 0.250 | 0.110 | 0.267 | 0.438 | 0.586 | 0.430 | 0.579 |
| ETTm2 | 96 | **0.059** | **0.176** | 0.074 | 0.206 | 0.067 | 0.198 | 0.065 | 0.189 | 0.088 | 0.225 | 0.075 | 0.208 |
| | 192 | 0.100 | 0.234 | **0.098** | **0.238** | 0.102 | 0.245 | 0.118 | 0.256 | 0.132 | 0.283 | 0.129 | 0.275 |
| | 336 | 0.153 | 0.301 | 0.135 | 0.282 | **0.130** | **0.279** | 0.154 | 0.305 | 0.180 | 0.336 | 0.154 | 0.302 |
| | 720 | 0.210 | 0.354 | **0.175** | **0.326** | 0.178 | 0.325 | 0.182 | 0.335 | 0.300 | 0.435 | 0.160 | 0.321 |

because of the lack of ability to capture complex temporal patterns of the time series. Meanwhile, MICN can achieve almost the same better performance when $i$ takes two or three values, indicating that the multi-branch structure is effective. To be more representative, we set $i$ to $\{12, 16\}$ in this paper.

## A.5 SELECTION OF DIFFERENT CONVOLUTION MODES

As shown in Table 11, we also record the performance in different convolution modes: $stride = kernel$ and $stride = \frac{kernel}{2}$. The second mode makes more comprehensive use of local information, making the convolution more coherent. MICN achieves similar performance in different convolution modes. It proves that MICN can make the most of sequence information, and the performance of the model depends on the structure we proposed.

Table 10: Multivariate results with different parameters *i* in three datasets: Electricity, Exchange and Traffic.

| Datasets | | Electricity | | | | Exchange | | | | Traffic | | | |
|---|---|---|---|---|---|---|---|---|---|---|---|---|---|
| Prediction Length O | | 96 | 192 | 336 | 720 | 96 | 192 | 336 | 720 | 96 | 192 | 336 | 720 |
| < 24 > | MSE | 0.174 | 0.200 | 0.207 | 0.240 | 0.093 | 0.181 | 0.271 | 0.762 | 0.575 | 0.569 | 0.581 | 0.607 |
| | MAE | 0.282 | 0.303 | 0.317 | 0.336 | 0.227 | 0.321 | 0.404 | 0.675 | 0.334 | 0.316 | 0.323 | 0.339 |
| < 48 > | MSE | 0.167 | 0.179 | 0.195 | 0.265 | 0.080 | 0.185 | 0.288 | 0.758 | 0.512 | 0.532 | 0.556 | 0.595 |
| | MAE | 0.278 | 0.287 | 0.303 | 0.361 | 0.204 | 0.316 | 0.412 | 0.671 | 0.296 | 0.304 | 0.315 | 0.330 |
| < 12, 16 > | MSE | 0.164 | **0.177** | 0.193 | 0.212 | 0.102 | 0.172 | 0.272 | 0.714 | 0.513 | 0.537 | 0.534 | 0.577 |
| | MAE | 0.269 | **0.285** | 0.304 | 0.321 | 0.235 | 0.316 | 0.407 | 0.658 | 0.309 | 0.315 | 0.313 | 0.325 |
| < 16, 24 > | MSE | **0.160** | 0.182 | **0.192** | 0.232 | 0.086 | 0.198 | **0.266** | **0.632** | 0.524 | 0.545 | 0.547 | 0.584 |
| | MAE | **0.267** | 0.291 | **0.299** | 0.341 | 0.210 | 0.334 | **0.388** | **0.639** | 0.300 | 0.310 | 0.317 | 0.329 |
| < 12, 24 > | MSE | 0.160 | 0.185 | 0.195 | 0.220 | 0.100 | **0.153** | 0.269 | 0.775 | 0.517 | 0.537 | 0.546 | 0.573 |
| | MAE | 0.268 | 0.293 | 0.307 | 0.329 | 0.231 | **0.295** | 0.403 | 0.678 | 0.303 | 0.310 | 0.319 | 0.319 |
| < 24, 48 > | MSE | 0.180 | 0.201 | 0.211 | 0.250 | **0.079** | 0.175 | 0.269 | 0.658 | 0.537 | 0.587 | 0.607 | 0.603 |
| | MAE | 0.288 | 0.306 | 0.316 | 0.344 | **0.203** | 0.310 | 0.401 | 0.634 | 0.307 | 0.324 | 0.329 | 0.340 |
| < 6, 12, 24 > | MSE | 0.169 | 0.180 | 0.195 | 0.215 | 0.114 | 0.208 | 0.299 | 0.798 | **0.513** | **0.522** | **0.535** | **0.560** |
| | MAE | 0.278 | 0.287 | 0.300 | 0.323 | 0.244 | 0.348 | 0.425 | 0.704 | **0.303** | **0.304** | **0.307** | **0.321** |
| < 12, 24, 48 > | MSE | 0.168 | 0.185 | 0.200 | **0.212** | 0.113 | 0.213 | 0.364 | 0.680 | 0.521 | 0.559 | 0.554 | 0.604 |
| | MAE | 0.274 | 0.293 | 0.305 | **0.320** | 0.247 | 0.354 | 0.462 | 0.655 | 0.305 | 0.317 | 0.317 | 0.336 |

Table 11: MICN performance under different convolution modes. We implement it on three multivariate datasets: Electricity, Exchange and Traffic.

| Datasets | | Electricity | | | | Exchange | | | | Traffic | | | |
|---|---|---|---|---|---|---|---|---|---|---|---|---|---|
| Prediction Length O | | 96 | 192 | 336 | 720 | 96 | 192 | 336 | 720 | 96 | 192 | 336 | 720 |
| *stride = kernel* | MSE | 0.164 | **0.177** | **0.193** | **0.212** | 0.102 | **0.172** | **0.272** | 0.714 | 0.519 | 0.537 | **0.534** | 0.577 |
| | MAE | 0.269 | 0.285 | **0.304** | **0.321** | 0.235 | 0.316 | **0.407** | 0.658 | 0.309 | 0.315 | 0.313 | 0.325 |
| $stride = \frac{kernel}{2}$ | MSE | **0.158** | 0.177 | 0.198 | 0.221 | **0.081** | 0.173 | 0.305 | **0.706** | **0.509** | **0.534** | 0.545 | **0.563** |
| | MAE | **0.267** | **0.283** | 0.310 | 0.326 | **0.208** | **0.314** | 0.430 | **0.647** | **0.306** | **0.303** | **0.310** | **0.323** |

# B  ADDITIONAL MODEL ANALYSIS

## B.1  MULTI-SCALE HYBRID DECOMPOSITION

Autoformer harnesses the decomposition as an inner block of deep models and gets good performance. However, the patterns obtained by its decomposition are simple and cannot effectively deal with the complex and changeable properties of time series. As shown in Table 12, we replace the decomposition block in Autoformer with our proposed multi-scale hybrid decomposition block. For Exchange, we achieve a similar performance because it has no obvious temporal pattern. The result verifies that multi-scale hybrid decomposition structure is more in line with the complex temporal patterns in real-time series.

## B.2  VISUALIZATION OF LEARNED TREND-CYCLICAL PARTS

As shown in Figure 6 and Figure 7, we plot the results of learned trend-cyclical parts. The separate modeling of the trend-cyclical part makes better performance and grasp of long-term progression. We also observe that the mean prediction is slightly better on the ETTm2 dataset. This is due to the complexity of the trend-cyclical information and the inability of simple linear regression, which may

Table 12: Ablation of multi-scale decomposition (MHDecomp). **Autoformer-MHDecomp** adopts multi-scale decomposition block into Autoformer.

| Datasets | | Electricity | | | | Exchange | | | | Traffic | | | |
|---|---|---|---|---|---|---|---|---|---|---|---|---|---|
| Prediction Length O | | 96 | 192 | 336 | 720 | 96 | 192 | 336 | 720 | 96 | 192 | 336 | 720 |
| Autoformer | MSE | 0.207 | 0.236 | 0.275 | **0.289** | **0.160** | 0.327 | **0.509** | **1.133** | 0.675 | **0.666** | 0.765 | 1.098 |
| | MAE | 0.323 | 0.343 | 0.372 | **0.380** | 0.292 | 0.415 | **0.527** | **0.825** | 0.406 | **0.425** | 0.487 | 0.647 |
| Autoformer-MHDecomp | MSE | **0.197** | 0.236 | **0.253** | 0.291 | 0.162 | **0.291** | 0.545 | 1.135 | **0.653** | 0.678 | **0.673** | **0.800** |
| | MAE | **0.312** | **0.340** | **0.356** | 0.382 | 0.292 | **0.392** | 0.552 | 0.826 | **0.402** | 0.427 | **0.421** | **0.493** |

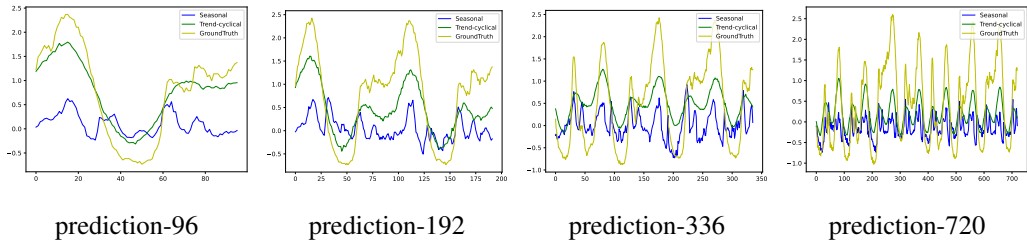

Figure 6: Visualization of learned trend-cyclical part prediction result $Y_t$ and seasonal part prediction result $Y_s$ in ETTm1 dataset under MICN-regre. Sample linear regression performs well.

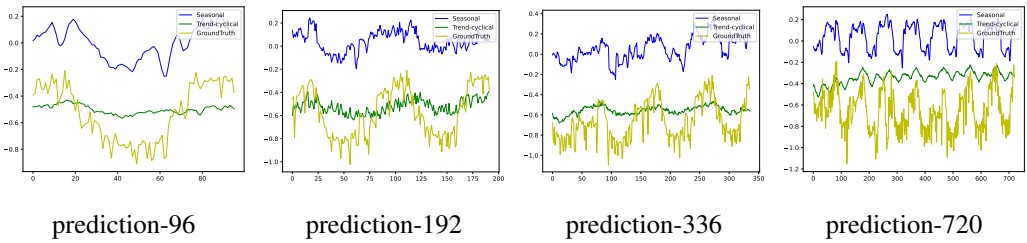

Figure 7: Visualization of $Y_t$ and $Y_s$ in ETTm2 dataset under MICN-regre. Sample linear regression does not perform very well.

require a more advanced trend prediction method. Moreover, the mean value of its trend change is close to constant, so mean prediction is better in this situation.

## B.3 ISOMETRIC CONVOLUTION VS. MASKED SELF-ATTENTION

With the local module in MICN, we get a short sequence characterizing local features. On this basis, we propose the isometric convolution in global module to model the global correlation of the sequence, while previously the first choice is masked self-attention. We replace the isometric convolution in the global module of MICN with masked self-attention for training, and the results are shown in Table 13 and Table 14. It verifies that for a short sequence, isometric convolution outperforms masked self-attention in general.

Table 13: Ablation of isometric convolution. We **replace the Isometric convolution in MICN-regre with masked self-attention** and implement it in the **multivariate** Electricity, Exchange and Traffic. The better results are highlighted in bold.

| Datasets | | Electricity | | | | Exchange | | | | Traffic | | | |
|---|---|---|---|---|---|---|---|---|---|---|---|---|---|
| Prediction Length $O$ | | 96 | 192 | 336 | 720 | 96 | 192 | 336 | 720 | 96 | 192 | 336 | 720 |
| Isometric | MSE | 0.164 | **0.177** | **0.193** | **0.212** | 0.102 | **0.172** | **0.272** | **0.714** | 0.519 | 0.537 | **0.534** | 0.577 |
| Convolution | MAE | 0.269 | **0.285** | **0.304** | **0.321** | 0.235 | **0.316** | **0.407** | **0.658** | 0.309 | 0.315 | 0.313 | **0.325** |
| self- | MSE | **0.153** | 0.179 | 0.201 | 0.256 | **0.096** | 0.209 | 0.311 | 0.960 | **0.501** | **0.522** | 0.543 | **0.568** |
| attention | MAE | **0.260** | 0.286 | 0.310 | 0.352 | **0.227** | 0.349 | 0.441 | 0.747 | **0.295** | **0.293** | **0.303** | 0.326 |

To compare the isometric convolution and the masked self-attention further, we conduct more experiments on full benchmark with different kernel sizes. The results in Table 15 show that the isometric convolution outperforms the masked self-attention in the most cases. And we also note that in some cases the masked self-attention is slightly more effective. We believe that this is related to the corresponding datasets and that we will analyse the datasets in detail in the future. Moreover, we can conclude that different kernels have a relatively small impact on the final results, which indicates that our advanced structure instead of model parameters plays a major role in the performance.

Table 14: Comparison of Isometric convolution and masked self-attention in the **univariate** Electricity, Exchange and Traffic. We **replace** the Isometric convolution in **MICN-regre** with masked self-attention. The better results are highlighted in bold.

| Datasets | | Electricity | | | | Exchange | | | | Traffic | | | |
|---|---|---|---|---|---|---|---|---|---|---|---|---|---|
| Prediction Length $O$ | | 96 | 192 | 336 | 720 | 96 | 192 | 336 | 720 | 96 | 192 | 336 | 720 |
| Isometric | MSE | **0.310** | **0.300** | **0.323** | **0.364** | **0.099** | **0.198** | **0.302** | 0.738 | 0.158 | 0.154 | 0.165 | 0.182 |
| Convolution | MAE | **0.398** | **0.394** | **0.413** | **0.449** | 0.240 | **0.354** | **0.447** | 0.662 | 0.241 | 0.236 | 0.243 | 0.264 |
| self- | MSE | 0.404 | 0.351 | 0.384 | 0.398 | 0.101 | 0.209 | 0.303 | **0.564** | **0.140** | **0.153** | **0.148** | **0.166** |
| attention | MAE | 0.461 | 0.428 | 0.456 | 0.467 | **0.237** | 0.369 | 0.450 | **0.600** | **0.217** | **0.233** | **0.228** | **0.249** |

Table 15: Comparison of the isometric convolution and the masked self-attention in MICN with **different kernel sizes**.

| Datasets | | | ETTm2 | | | | Electricity | | | | Exchange | | | | Traffic | | | | WTH | | | |
|---|---|---|---|---|---|---|---|---|---|---|---|---|---|---|---|---|---|---|---|---|---|---|---|
| Prediction Length $O$ | | | 96 | 192 | 336 | 720 | 96 | 192 | 336 | 720 | 96 | 192 | 336 | 720 | 96 | 192 | 336 | 720 | 96 | 192 | 336 | 720 |
| <12,16> | Isometric convolution | MSE | 0.179 | 0.307 | 0.325 | 0.502 | 0.164 | 0.177 | 0.193 | 0.212 | 0.102 | 0.172 | 0.272 | 0.714 | 0.519 | 0.537 | 0.534 | 0.577 | 0.161 | 0.220 | 0.278 | 0.311 |
| | | MAE | 0.275 | 0.376 | 0.388 | 0.490 | 0.269 | 0.285 | 0.304 | 0.321 | 0.235 | 0.316 | 0.407 | 0.658 | 0.309 | 0.315 | 0.313 | 0.325 | 0.229 | 0.281 | 0.331 | 0.356 |
| | self-attention | MSE | 0.186 | 0.292 | 0.448 | 0.520 | 0.153 | 0.179 | 0.201 | 0.256 | 0.096 | 0.209 | 0.311 | 0.960 | 0.501 | 0.522 | 0.543 | 0.568 | 0.170 | 0.220 | 0.266 | 0.331 |
| | | MAE | 0.287 | 0.373 | 0.478 | 0.527 | 0.260 | 0.286 | 0.310 | 0.352 | 0.227 | 0.349 | 0.441 | 0.747 | 0.295 | 0.293 | 0.303 | 0.326 | 0.242 | 0.282 | 0.323 | 0.375 |
| <11,19> | Isometric convolution | MSE | 0.183 | 0.275 | 0.308 | 0.415 | 0.164 | 0.178 | 0.187 | 0.213 | 0.091 | 0.175 | 0.300 | 0.767 | 0.519 | 0.540 | 0.531 | 0.582 | 0.164 | 0.213 | 0.258 | 0.326 |
| | | MAE | 0.277 | 0.349 | 0.364 | 0.429 | 0.271 | 0.286 | 0.297 | 0.322 | 0.226 | 0.320 | 0.413 | 0.679 | 0.306 | 0.314 | 0.309 | 0.319 | 0.229 | 0.271 | 0.315 | 0.366 |
| | self-attention | MSE | 0.193 | 0.326 | 0.335 | 0.445 | 0.170 | 0.176 | 0.205 | 0.268 | 0.092 | 0.205 | 0.330 | 0.997 | 0.504 | 0.519 | 0.539 | 0.565 | 0.169 | 0.214 | 0.274 | 0.326 |
| | | MAE | 0.281 | 0.362 | 0.388 | 0.455 | 0.274 | 0.283 | 0.309 | 0.355 | 0.222 | 0.340 | 0.455 | 0.754 | 0.295 | 0.304 | 0.315 | 0.329 | 0.233 | 0.278 | 0.334 | 0.369 |
| <13,25> | Isometric convolution | MSE | 0.175 | 0.242 | 0.313 | 0.499 | 0.167 | 0.180 | 0.191 | 0.217 | 0.083 | 0.177 | 0.279 | 0.726 | 0.517 | 0.542 | 0.545 | 0.572 | 0.163 | 0.213 | 0.271 | 0.321 |
| | | MAE | 0.268 | 0.319 | 0.373 | 0.486 | 0.273 | 0.288 | 0.300 | 0.323 | 0.213 | 0.317 | 0.403 | 0.665 | 0.304 | 0.315 | 0.319 | 0.324 | 0.234 | 0.272 | 0.335 | 0.357 |
| | self-attention | MSE | 0.214 | 0.314 | 0.301 | 0.464 | 0.164 | 0.175 | 0.187 | 0.221 | 0.098 | 0.174 | 0.345 | 1.266 | 0.523 | 0.532 | 0.560 | 0.571 | 0.162 | 0.221 | 0.276 | 0.317 |
| | | MAE | 0.294 | 0.382 | 0.357 | 0.467 | 0.268 | 0.281 | 0.295 | 0.327 | 0.226 | 0.302 | 0.455 | 0.839 | 0.309 | 0.309 | 0.324 | 0.329 | 0.234 | 0.286 | 0.338 | 0.359 |
| <15,23> | Isometric convolution | MSE | 0.174 | 0.266 | 0.295 | 0.439 | 0.163 | 0.177 | 0.197 | 0.208 | 0.086 | 0.153 | 0.291 | 0.687 | 0.522 | 0.531 | 0.552 | 0.584 | 0.175 | 0.214 | 0.278 | 0.323 |
| | | MAE | 0.269 | 0.347 | 0.353 | 0.447 | 0.268 | 0.286 | 0.309 | 0.317 | 0.212 | 0.287 | 0.419 | 0.638 | 0.295 | 0.312 | 0.316 | 0.332 | 0.243 | 0.278 | 0.334 | 0.366 |
| | self-attention | MSE | 0.184 | 0.300 | 0.301 | 0.467 | 0.157 | 0.177 | 0.185 | 0.228 | 0.100 | 0.167 | 0.336 | 1.114 | 0.490 | 0.518 | 0.542 | 0.571 | 0.167 | 0.221 | 0.268 | 0.317 |
| | | MAE | 0.283 | 0.369 | 0.358 | 0.466 | 0.262 | 0.284 | 0.292 | 0.300 | 0.231 | 0.300 | 0.444 | 0.804 | 0.287 | 0.301 | 0.304 | 0.334 | 0.236 | 0.285 | 0.328 | 0.357 |

## B.4 COMPARISON OF MERGING OPERATIONS

The traditional method of merging branch structures is the concat operation on the hidden state. In this paper, we propose to adopt 2D convolution to merge multiple branches to better measure the importance of each branch (the kernel represents the weights). As shown in Table 16, the better performance verifies the effectiveness of our proposed method.

Table 16: Comparison of different merging operations. The better results are highlighted in bold.

| Datasets | | Electricity | | | | Exchange | | | | Traffic | | | |
|---|---|---|---|---|---|---|---|---|---|---|---|---|---|
| Prediction Length $O$ | | 96 | 192 | 336 | 720 | 96 | 192 | 336 | 720 | 96 | 192 | 336 | 720 |
| MICN | MSE | 0.164 | **0.177** | **0.193** | **0.212** | 0.102 | **0.172** | 0.272 | **0.714** | **0.519** | **0.537** | **0.534** | **0.577** |
| *Conv2d* | MAE | 0.269 | **0.285** | **0.304** | **0.321** | 0.235 | **0.316** | 0.407 | **0.658** | **0.309** | **0.315** | **0.313** | **0.325** |
| MICN- | MSE | **0.160** | 0.183 | 0.202 | 0.214 | **0.099** | 0.175 | **0.264** | 0.711 | 0.528 | 0.549 | 0.536 | 0.577 |
| *concat* | MAE | **0.266** | 0.289 | 0.310 | 0.324 | **0.229** | 0.322 | **0.403** | 0.655 | 0.313 | 0.318 | 0.316 | 0.326 |

## B.5 EFFICIENCY ANALYSIS

For MICN, the complexity lies in Downsampling convolution and Isometric convolution in Local-Global module. If we set the sequence length to $L$, the hidden state to $D$ and the multi-scale convolution kernels to $i$.

For downsampling convolution (kernel= stride), the complexity is $\mathscr{O}(i * D^2 * \frac{L}{i}) = \mathscr{O}(LD^2)$.

For Isometric convolution, the sequence length and kernel are changed to $\frac{L}{i}$, stride=1, padding=$\frac{L}{i} - 1$, so the complexity is $\mathscr{O}((\frac{L}{i})^2 * D^2) = \mathscr{O}(\frac{L^2 D^2}{i^2})$. In this paper, we set $i \in \{\frac{L}{4}, \frac{L}{8}...\}$ is factor of $L$, so the complexity of Isometric convolution is $\mathscr{O}(cD^2)$, where c is a constant.

In summary, the overall complexity is $max(\mathscr{O}(LD^2), \mathscr{O}(cD^2)) = \mathscr{O}(LD^2)$, which is linear about the sequence length. The comparisons of the time complexity and memory usage in training and the inference steps in testing are summarized in Table 17.

Furthermore, we compare the running memory and time among Local-Global-based, Auto-correlation-based and self-attention-based models during the training phase. As shown in Figure 8, the proposed Local-Global module shows $\mathscr{O}(LD^2)$ complexity and achieves better long-term sequences efficiency.

Table 17: Complexity analysis of different forecasting models.

| Methods | Training | |
|---|---|---|
| | Time | Memory |
| MICN | $\mathcal{O}(LD^2)$ | $\mathcal{O}(LD^2)$ |
| FEDformer (Zhou et al., 2022) | $\mathcal{O}(L)$ | $\mathcal{O}(L)$ |
| Autoformer (Wu et al., 2021b) | $\mathcal{O}(L\log L)$ | $\mathcal{O}(L\log L)$ |
| Informer (Zhou et al., 2021) | $\mathcal{O}(L\log L)$ | $\mathcal{O}(L\log L)$ |
| LogTrans (Li et al., 2019b) | $\mathcal{O}(L\log L)$ | $\mathcal{O}\left(L^2\right)$ |
| Transformer (Vaswani et al., 2017) | $\mathcal{O}\left(L^2\right)$ | $\mathcal{O}\left(L^2\right)$ |
| LSTM (Hochreiter & Schmidhuber, 1997) | $\mathcal{O}(L)$ | $\mathcal{O}(L)$ |

As the prediction length increases, our model takes a little more time than Auto-Correlation. We speculate that this may be due to the use of the convolution operation or the activation function *Tanh*. In general, our method is the most portable and valuable in practical applications.

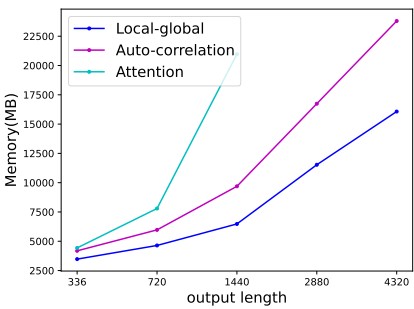 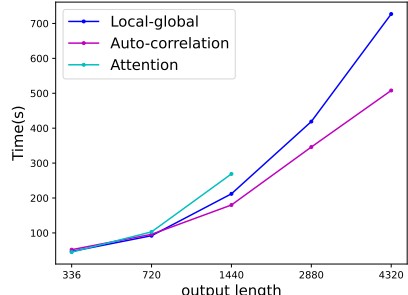

Memory Efficiency Analysis        Running Time Efficiency Analysis

Figure 8: Efficiency Analysis. We place the Local-Global module in MICN with Auto-correlation and self-attention. Then we record the memory and running time of an epoch with fixed input length 96 and increasing output length. Missing values of self-attention are due to out-of-memory.

## B.6 MORE ANALYSIS OF THE TREND-CYCLICAL PREDICTION BLOCK

To further evaluate the trend-cyclical prediction, we conduct more experiments on full benchmark. As shown in Table 18, **regre** is the simple regression prediction without other modules and **mean** is the simple mean prediction without other modules. **MICN-regre** is our proposed method with regression prediction in trend-cyclical block and **MICN-mean** is the mean prediction in trend-cyclical block. **FEDformer-mean** is the original FEDformer using the mean prediction and **FEDformer-regre** uses regression prediction instead of mean prediction. We can conclude that simple regression and mean fail to capture the complex temporal correlations. But for the data Exchange without periodicity, simple regression makes competitive performance. This result is worth thinking about, and we will conduct more in-depth experiments in the future. Meanwhile, the results that MICN-regre outperforms FEDformer-regre and MICN-mean outperforms FEDforemr-mean prove the validity of our proposed model. We also find FEDformer-regre makes worse performance in most cases. This may be due to the more complex structure of FEDformer, and more complex regression prediction is needed correspondingly.

## B.7 THE DETAILED DESCRIPTION OF THE LOCAL-GLOBAL MODULE

We show the detailed description of the Local-global module in Figure 9. For the input series, we adopt down-sampling convolution with different kernels to extract the local features of different temporal patterns and isometric convolution instead of masked self-attention to capture global correlations. Then we use up-sampling convolution to recover the length of the series. Finally, we merge the different branches to complete the modeling of different patterns.

Table 18: Comparison of regression prediction and mean prediction with different models. The better results are highlighted in bold.

| Datasets | | ETTm2 | | | | Electricity | | | | Exchange | | | | Traffic | | | | WTH | | | |
|---|---|---|---|---|---|---|---|---|---|---|---|---|---|---|---|---|---|---|---|---|---|
| Prediction Length $O$ | | 96 | 192 | 336 | 720 | 96 | 192 | 336 | 720 | 96 | 192 | 336 | 720 | 96 | 192 | 336 | 720 | 96 | 192 | 336 | 720 |
| regre | MSE | 0.187 | 0.267 | 0.331 | 0.564 | 0.216 | 0.216 | 0.229 | 0.263 | **0.087** | **0.165** | **0.255** | 0.748 | 0.707 | 0.658 | 0.662 | 0.704 | 0.476 | 0.696 | 1.010 | 1.594 |
| | MAE | 0.286 | 0.350 | 0.392 | 0.541 | 0.311 | 0.314 | 0.328 | 0.358 | **0.209** | **0.304** | **0.389** | 0.678 | 0.440 | 0.419 | 0.421 | 0.440 | 0.526 | 0.649 | 0.821 | 0.993 |
| MICN - regre | MSE | **0.179** | 0.307 | 0.325 | 0.502 | **0.164** | **0.177** | **0.193** | **0.212** | 0.102 | 0.172 | 0.272 | **0.714** | **0.519** | **0.537** | **0.534** | **0.577** | **0.161** | **0.220** | **0.278** | **0.311** |
| | MAE | **0.275** | 0.376 | 0.388 | 0.490 | **0.269** | **0.285** | **0.304** | **0.321** | 0.235 | 0.316 | 0.407 | **0.658** | **0.309** | **0.315** | **0.313** | **0.325** | 0.229 | 0.281 | **0.331** | **0.356** |
| FEDformer - regre | MSE | 0.238 | 0.385 | 0.491 | 0.691 | 0.195 | 0.212 | 0.225 | 0.261 | 0.126 | 0.210 | 0.342 | 0.795 | 0.589 | 0.614 | 0.635 | 0.652 | 0.287 | 0.387 | 0.662 | 0.755 |
| | MAE | 0.342 | 0.444 | 0.506 | 0.602 | 0.308 | 0.323 | 0.337 | 0.364 | 0.269 | 0.355 | 0.459 | 0.696 | 0.350 | 0.366 | 0.370 | 0.371 | 0.378 | 0.441 | 0.591 | 0.664 |
| mean | MSE | 0.216 | 0.275 | 0.334 | 0.427 | 0.523 | 0.532 | 0.543 | 0.575 | 0.160 | 0.282 | 0.454 | 1.302 | 1.196 | 1.206 | 1.218 | 1.244 | 0.228 | 0.278 | 0.325 | 0.394 |
| | MAE | 0.298 | 0.332 | 0.370 | 0.424 | 0.556 | 0.561 | 0.567 | 0.575 | 0.289 | 0.384 | 0.493 | 0.881 | 0.698 | 0.701 | 0.706 | 0.709 | 0.293 | 0.331 | 0.353 | 0.392 |
| MICN - mean | MSE | 0.200 | **0.262** | **0.305** | **0.389** | 0.188 | 0.200 | 0.219 | 0.224 | 0.173 | 0.324 | 0.639 | 1.218 | 0.575 | 0.580 | 0.583 | 0.601 | 0.183 | 0.246 | 0.293 | 0.373 |
| | MAE | 0.287 | **0.326** | **0.353** | **0.407** | 0.302 | 0.308 | 0.328 | 0.332 | 0.297 | 0.408 | 0.598 | 0.862 | 0.344 | 0.349 | 0.345 | 0.363 | 0.250 | 0.317 | 0.335 | 0.399 |
| FEDformer - mean | MSE | 0.192 | 0.254 | 0.326 | 0.433 | 0.188 | 0.197 | 0.214 | 0.244 | 0.139 | 0.266 | 0.457 | 1.152 | 0.575 | 0.611 | 0.618 | 0.630 | 0.209 | 0.273 | 0.331 | 0.409 |
| | MAE | 0.283 | 0.322 | 0.366 | 0.423 | 0.303 | 0.311 | 0.328 | 0.353 | 0.269 | 0.376 | 0.498 | 0.822 | 0.357 | 0.378 | 0.380 | 0.383 | 0.291 | 0.333 | 0.374 | 0.418 |

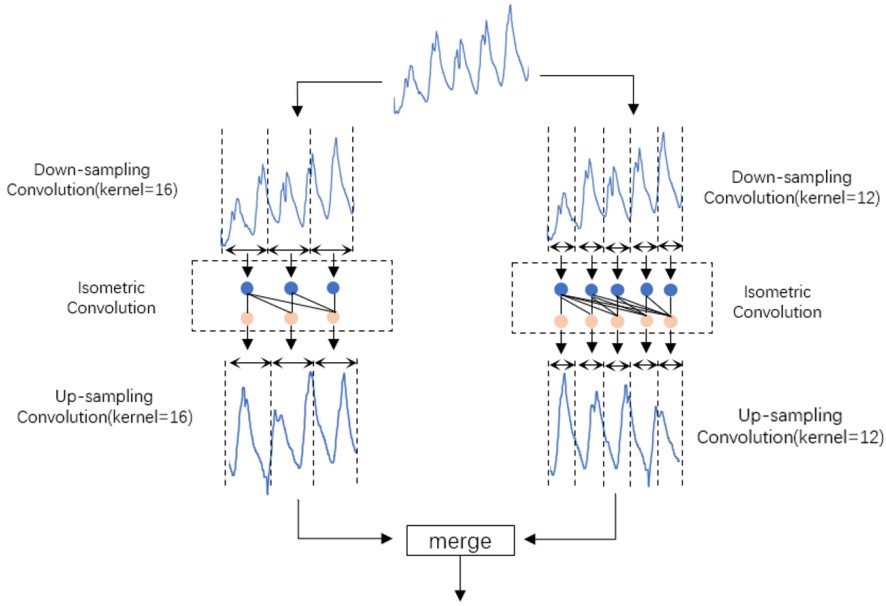

Figure 9: The detail of the local-global module.

## B.8 MORE DISCUSSION ABOUT THE ISOMETRIC CONVOLUTION AND THE LINEAR COMPLEXITY MODEL: LINFORMER, FASTFORMER.

Linformer Wang et al. (2020) and Fastformer Wu et al. (2021a) are other advanced self-attention family models with linear complexity. We conduct more comparison experiments on full benchmark and show the results in Table 19 and Table 20. Concretely, we replace the isometric convolution in MICN-regre with the different masked attention mechanisms in Linformer and Fastformer. We implement the experiments in multivariate and univariate datasets. We can conclude that Fastformer can make relatively competitive performance but our proposed model MICN performs best in general. These results provide further evidence of the effectiveness of MICN.

Table 19: Comparison of different linear complexity models on the multivariate datasets. The better results are highlighted in bold.

| Datasets | | ETTm2 | | | | Electricity | | | | Exchange | | | | Traffic | | | | WTH | | | |
|---|---|---|---|---|---|---|---|---|---|---|---|---|---|---|---|---|---|---|---|---|---|
| Prediction Length $O$ | | 96 | 192 | 336 | 720 | 96 | 192 | 336 | 720 | 96 | 192 | 336 | 720 | 96 | 192 | 336 | 720 | 96 | 192 | 336 | 720 |
| MICN- regre | MSE | **0.179** | 0.307 | **0.325** | 0.502 | **0.164** | **0.177** | **0.193** | **0.212** | 0.102 | 0.172 | **0.272** | 0.714 | **0.519** | **0.537** | **0.534** | **0.577** | **0.161** | **0.220** | 0.278 | 0.311 |
| | MAE | **0.275** | 0.376 | **0.388** | 0.490 | **0.269** | **0.285** | **0.304** | **0.321** | 0.235 | 0.316 | **0.407** | 0.658 | **0.309** | **0.315** | **0.313** | **0.325** | 0.229 | **0.281** | **0.331** | **0.356** |
| Linformer- regre | MSE | 0.186 | **0.269** | 0.369 | **0.451** | 0.190 | 0.195 | 0.208 | 0.239 | **0.083** | **0.166** | 0.286 | 0.850 | 0.556 | 0.540 | 0.551 | 0.581 | 0.200 | 0.240 | 0.291 | 0.398 |
| | MAE | 0.277 | **0.339** | 0.418 | **0.461** | 0.283 | 0.290 | 0.305 | 0.332 | **0.207** | **0.306** | 0.415 | 0.713 | 0.337 | 0.332 | 0.337 | 0.346 | 0.266 | 0.303 | 0.343 | 0.438 |
| Fastformer- regre | MSE | 0.192 | 0.329 | 0.394 | 0.563 | 0.167 | 0.189 | 0.216 | 0.237 | 0.116 | 0.170 | 0.309 | **0.709** | 0.531 | 0.556 | 0.552 | 0.591 | 0.179 | 0.225 | **0.270** | 0.338 |
| | MAE | 0.293 | 0.378 | 0.436 | 0.527 | 0.275 | 0.296 | 0.319 | 0.338 | 0.250 | 0.315 | 0.433 | 0.670 | 0.314 | 0.328 | 0.323 | 0.340 | 0.248 | 0.297 | **0.325** | 0.377 |

Table 20: Comparison of different linear complexity models on the univariate datasets. The better results are highlighted in bold.

| Datasets | | ETTm2 | | | | Electricity | | | | Exchange | | | | Traffic | | | | WTH | | | |
|---|---|---|---|---|---|---|---|---|---|---|---|---|---|---|---|---|---|---|---|---|---|
| Prediction Length O | | 96 | 192 | 336 | 720 | 96 | 192 | 336 | 720 | 96 | 192 | 336 | 720 | 96 | 192 | 336 | 720 | 96 | 192 | 336 | 720 |
| MICN-regre | MSE | **0.059** | 0.100 | 0.153 | 0.210 | **0.310** | **0.300** | **0.323** | **0.364** | **0.099** | 0.198 | **0.302** | 0.738 | **0.158** | **0.154** | **0.165** | **0.182** | **0.0029** | **0.0021** | **0.0023** | 0.0048 |
| | MAE | **0.176** | 0.234 | 0.301 | 0.354 | **0.398** | **0.394** | **0.413** | **0.449** | **0.240** | 0.354 | **0.447** | 0.662 | **0.241** | **0.236** | **0.243** | **0.264** | **0.039** | **0.034** | **0.034** | 0.054 |
| Linformer-regre | MSE | 0.079 | 0.131 | 0.157 | 0.225 | 0.361 | 0.384 | 0.406 | 0.486 | 0.113 | 0.203 | 0.353 | 0.746 | 0.172 | 0.206 | 0.211 | 0.233 | 0.0030 | 0.0035 | 0.0043 | 0.0046 |
| | MAE | 0.209 | 0.278 | 0.306 | 0.367 | 0.448 | 0.459 | 0.474 | 0.525 | 0.263 | 0.355 | 0.469 | 0.675 | 0.264 | 0.299 | 0.307 | 0.323 | 0.040 | 0.044 | 0.050 | 0.053 |
| Fastformer-regre | MSE | 0.063 | **0.092** | **0.121** | **0.190** | 0.408 | 0.338 | 0.370 | 0.393 | 0.120 | **0.185** | 0.352 | **0.607** | 0.176 | 0.184 | 0.183 | 0.201 | 0.0035 | 0.0035 | 0.0046 | **0.0041** |
| | MAE | 0.178 | **0.225** | **0.262** | **0.333** | 0.482 | 0.434 | 0.456 | 0.470 | 0.281 | **0.344** | 0.471 | **0.615** | 0.263 | 0.267 | 0.270 | 0.287 | 0.044 | 0.045 | 0.053 | **0.049** |

# C SUPPLEMENTARY OF MAIN RESULTS

## C.1 MAIN RESULTS WITH STANDARD DEVIATIONS

To get more robust experimental results, we repeat each experiment three times with different random seeds. For easier comparison, the results are shown in the main text when the seed is set to 2021. Table 21 shows the standard deviations.

Table 21: Quantitative results with fluctuations under different prediction lengths $O$ for multivariate forecasting. A lower MSE or MAE indicates a better performance.

| Methods | | MICN-regre | | MICN-mean | | Autoformer | | Informer | | LogTrans | |
|---|---|---|---|---|---|---|---|---|---|---|---|
| Metric | | MSE | MAE | MSE | MAE | MSE | MAE | MSE | MAE | MSE | MAE |
| ETTm2 | 96 | **0.177**±0.004 | **0.273**±0.004 | 0.204±0.003 | 0.289±0.002 | 0.255±0.020 | 0.339±0.020 | 0.365±0.062 | 0.453±0.047 | 0.768±0.071 | 0.642±0.020 |
| | 192 | 0.289±0.013 | 0.360±0.012 | **0.257**±0.004 | **0.323**±0.002 | 0.281±0.027 | 0.340±0.025 | 0.533±0.109 | 0.563±0.050 | 0.989±0.124 | 0.757±0.049 |
| | 336 | 0.324±0.001 | 0.381±0.006 | **0.310**±0.003 | **0.357**±0.004 | 0.339±0.018 | 0.372±0.015 | 1.363±0.173 | 0.887±0.056 | 1.334±0.168 | 0.872±0.054 |
| | 720 | 0.470±0.032 | 0.468±0.019 | **0.392**±0.008 | **0.407**±0.001 | 0.422±0.015 | 0.419±0.010 | 3.379±0.143 | 1.388±0.037 | 3.048±0.140 | 1.328±0.023 |
| Electricity | 96 | **0.163**±0.003 | **0.269**±0.002 | 0.190±0.005 | 0.303±0.004 | 0.201±0.003 | 0.317±0.004 | 0.274±0.004 | 0.368±0.003 | 0.258±0.002 | 0.357±0.002 |
| | 192 | **0.180**±0.002 | **0.288**±0.002 | 0.204±0.008 | 0.311±0.008 | 0.222±0.003 | 0.334±0.004 | 0.296±0.009 | 0.386±0.007 | 0.266±0.005 | 0.368±0.004 |
| | 336 | **0.193**±0.003 | **0.302**±0.002 | 0.218±0.102 | 0.326±0.007 | 0.231±0.006 | 0.338±0.004 | 0.300±0.007 | 0.394±0.004 | 0.280±0.006 | 0.380±0.001 |
| | 720 | **0.221**±0.012 | **0.326**±0.006 | 0.230±0.009 | 0.337±0.009 | 0.254±0.007 | 0.361±0.008 | 0.373±0.034 | 0.439±0.024 | 0.283±0.003 | 0.376±0.002 |
| Exchange | 96 | **0.093**±0.007 | **0.221**±0.010 | 0.172±0.007 | 0.299±0.002 | 0.197±0.019 | 0.323±0.012 | 0.847±0.150 | 0.752±0.060 | 0.968±0.177 | 0.812±0.027 |
| | 192 | **0.168**±0.003 | **0.314**±0.003 | 0.286±0.027 | 0.385±0.017 | 0.300±0.020 | 0.369±0.016 | 1.204±0.149 | 0.895±0.061 | 1.040±0.232 | 0.851±0.029 |
| | 336 | **0.269**±0.008 | **0.397**±0.009 | 0.552±0.064 | 0.552±0.035 | 0.509±0.041 | 0.524±0.016 | 1.672±0.036 | 1.036±0.014 | 1.659±0.122 | 1.081±0.015 |
| | 720 | **0.715**±0.022 | **0.666**±0.011 | 1.203±0.026 | 0.848±0.015 | 1.447±0.084 | 0.941±0.028 | 2.478±0.198 | 1.310±0.070 | 1.941±0.327 | 1.127±0.030 |
| Traffic | 96 | **0.521**±0.005 | **0.308**±0.002 | 0.575±0.002 | 0.347±0.005 | 0.613±0.028 | 0.388±0.012 | 0.719±0.015 | 0.391±0.004 | 0.684±0.041 | 0.384±0.008 |
| | 192 | **0.537**±0.008 | **0.313**±0.001 | 0.577±0.005 | 0.345±0.005 | 0.616±0.042 | 0.382±0.020 | 0.696±0.050 | 0.379±0.023 | 0.685±0.055 | 0.390±0.021 |
| | 336 | **0.536**±0.003 | **0.314**±0.001 | 0.587±0.005 | 0.350±0.005 | 0.622±0.016 | 0.337±0.011 | 0.777±0.009 | 0.420±0.003 | 0.733±0.069 | 0.408±0.026 |
| | 720 | **0.595**±0.014 | **0.325**±0.003 | 0.601±0.005 | 0.359±0.004 | 0.660±0.025 | 0.408±0.015 | 0.864±0.026 | 0.472±0.015 | 0.717±0.030 | 0.396±0.010 |
| Weather | 96 | **0.163**±0.003 | **0.231**±0.004 | 0.185±0.003 | 0.258±0.007 | 0.266±0.007 | 0.336±0.006 | 0.300±0.013 | 0.384±0.013 | 0.458±0.143 | 0.490±0.038 |
| | 192 | **0.216**±0.003 | **0.279**±0.001 | 0.239±0.005 | 0.308±0.007 | 0.307±0.024 | 0.367±0.022 | 0.598±0.045 | 0.544±0.028 | 0.658±0.151 | 0.589±0.032 |
| | 336 | **0.268**±0.010 | **0.321**±0.010 | 0.303±0.015 | 0.351±0.018 | 0.359±0.035 | 0.395±0.031 | 0.578±0.024 | 0.523±0.016 | 0.797±0.034 | 0.652±0.019 |
| | 720 | **0.319**±0.006 | **0.362**±0.005 | 0.355±0.030 | 0.400±0.005 | 0.419±0.017 | 0.428±0.014 | 1.059±0.096 | 0.741±0.042 | 0.869±0.045 | 0.675±0.093 |

## C.2 UNIVARIATE SHOWCASES

As shown in Figure 10, Figure 11, Figure 12, Figure 13, Figure 14, and Figure 15, we plot the forecasting results from the test set of univariate dataset Electricity and Traffic for comparison. Our model gives the best performance among different models. Moreover, MICN is significantly better at predicting the overall change and peak in the time series than Transformer-based models.

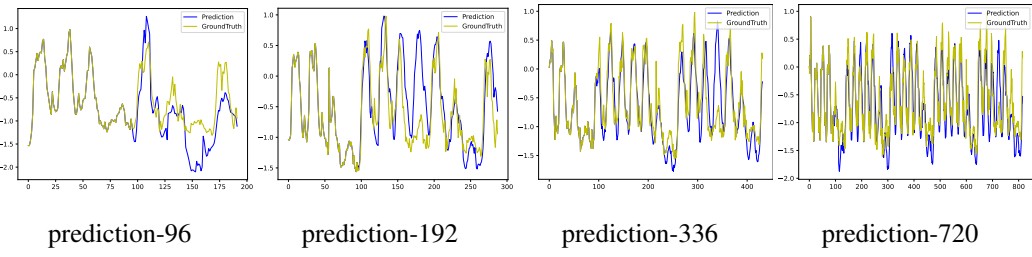

prediction-96  prediction-192  prediction-336  prediction-720

Figure 10: Prediction cases from the univariate Electricity dataset under MICN.

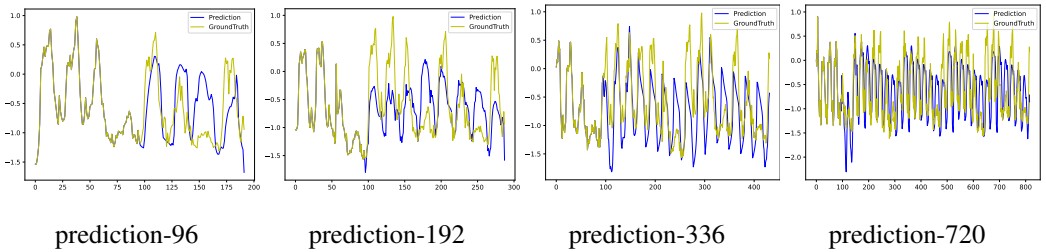

prediction-96      prediction-192      prediction-336      prediction-720

Figure 11: Prediction cases from the univariate Electricity dataset under Autoformer.

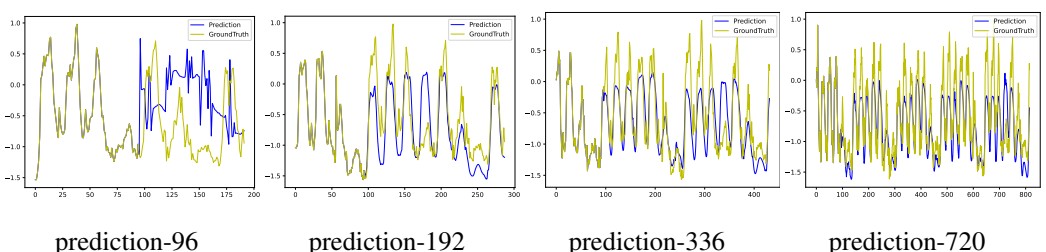

prediction-96      prediction-192      prediction-336      prediction-720

Figure 12: Prediction cases from the univariate Electricity dataset under Informer.

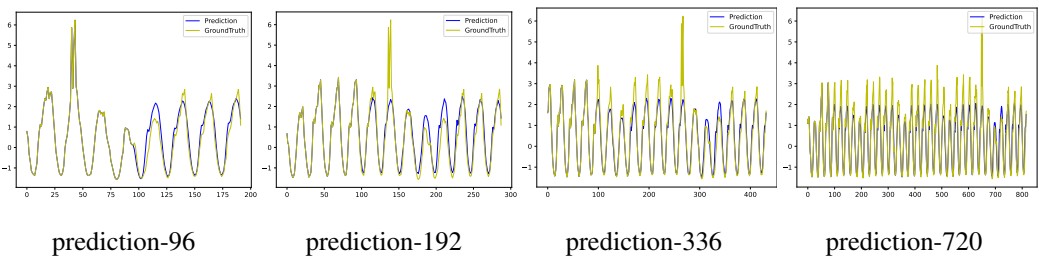

prediction-96      prediction-192      prediction-336      prediction-720

Figure 13: Prediction cases from the univariate Traffic dataset under MICN.

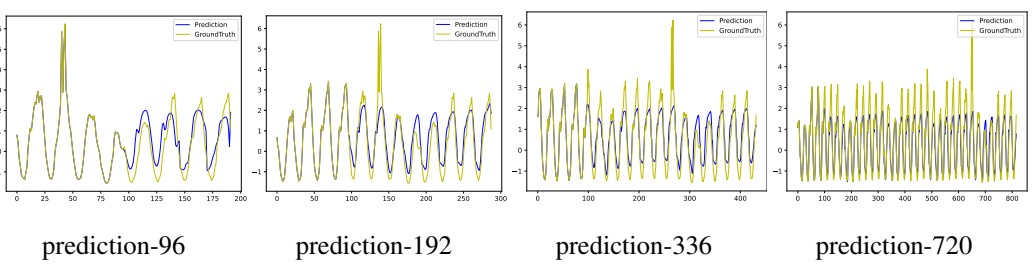

prediction-96      prediction-192      prediction-336      prediction-720

Figure 14: Prediction cases from the univariate Traffic dataset under Autoformer.

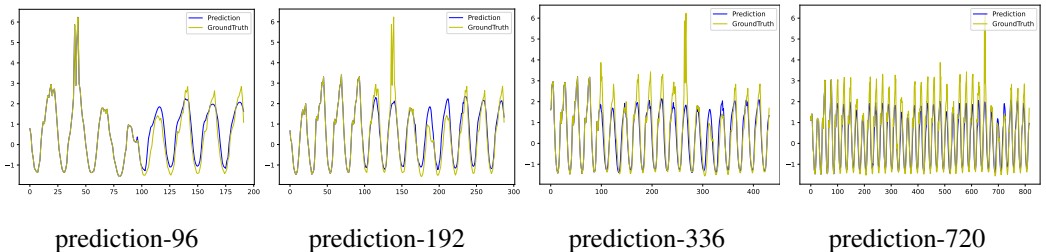

prediction-96     prediction-192     prediction-336     prediction-720

Figure 15: Prediction cases from the univariate Traffic dataset under Informer.

## C.3 MULTIVARIATE SHOWCASES

As shown in Figure 16, Figure 17, Figure 18, Figure 19, Figure 20, and Figure 21, we also plot the forecasting results from the test set of multivariate datasets ETTm1 and ETTm2 for comparison. Our model gives the most accurate prediction. Moreover, MICN is better at predicting rising and falling turning points in time series and closer to ground truth.

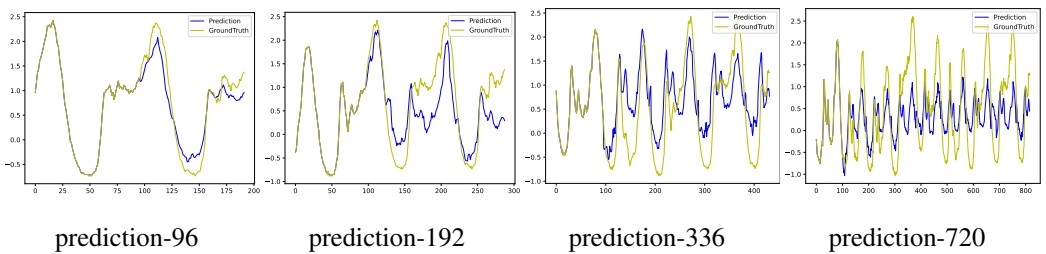

prediction-96     prediction-192     prediction-336     prediction-720

Figure 16: Prediction cases from the multivariate ETTm1 dataset under MICN.

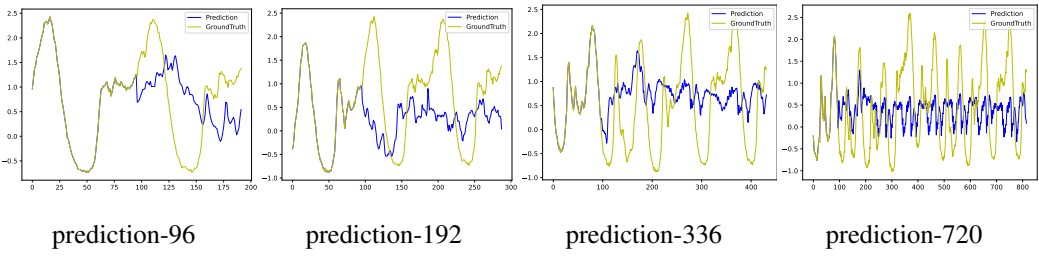

prediction-96     prediction-192     prediction-336     prediction-720

Figure 17: Prediction cases from the multivariate ETTm1 dataset under Autoformer.

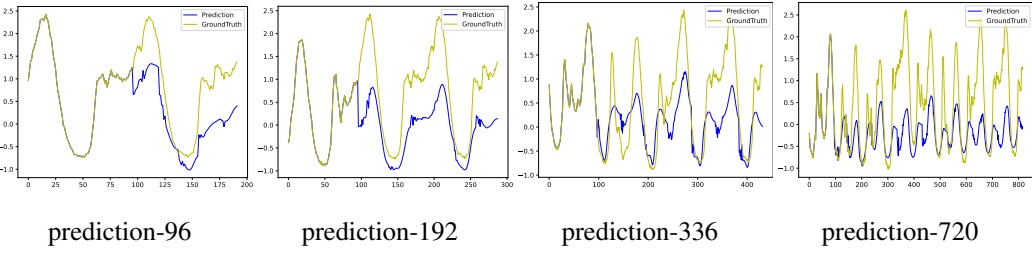

prediction-96     prediction-192     prediction-336     prediction-720

Figure 18: Prediction cases from the multivariate ETTm1 dataset under Informer.

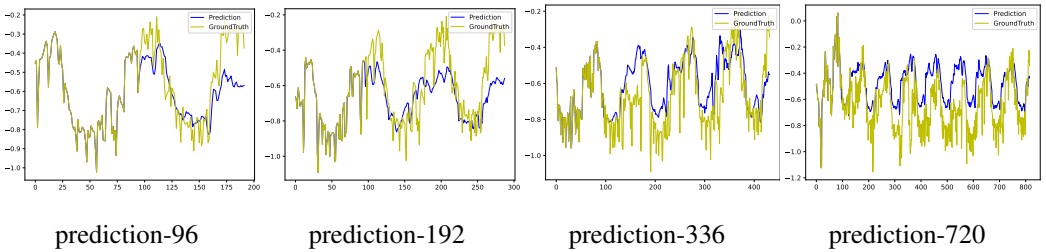

prediction-96          prediction-192          prediction-336          prediction-720

Figure 19: Prediction cases from the multivariate ETTm2 dataset under MICN.

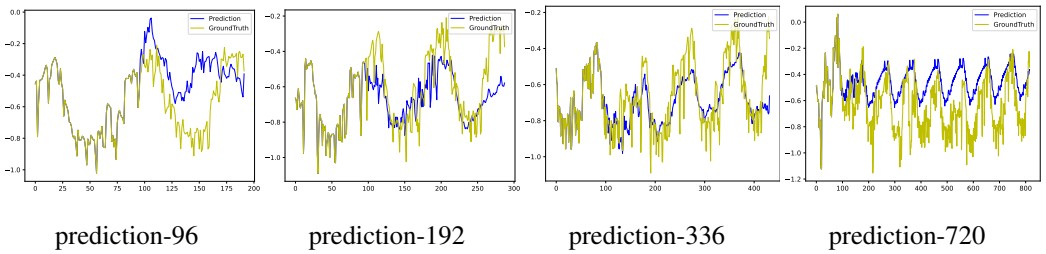

prediction-96          prediction-192          prediction-336          prediction-720

Figure 20: Prediction cases from the multivariate ETTm2 dataset under Autoformer.

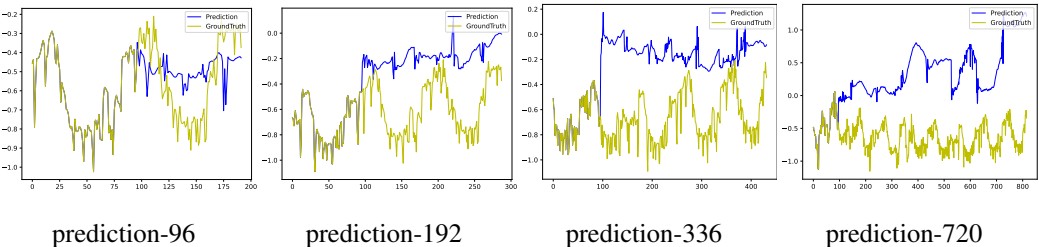

prediction-96          prediction-192          prediction-336          prediction-720

Figure 21: Prediction cases from the multivariate ETTm2 dataset under Informer.

