# OpenReview forum: "MICN: Multi-scale Local and Global Context Modeling for Long-term Series Forecasting"
_ICLR.cc/2023/Conference — ICLR 2023 notable top 5%_

### Official Review · Reviewer_PRkd · 2022-10-23

**Confidence:** 4
**Clarity, Quality, Novelty And Reproducibility:** Good
**Correctness:** 2
**Technical Novelty And Significance:** 3
**Empirical Novelty And Significance:** 3
**Recommendation:** 6

**Strength And Weaknesses:**

Pros:
- The results show that MICN achieves state-of-the-art accuracy on five real-world benchmarks. MICN yields impressive 18.2% and 24.5% relative improvements for multivariate and univariate time series, respectively.
- This paper uses a lot of ablation experiments to verify that the proposed Multi-scale Isometric Convolution outperforms the self-attention family and Auto-correlation mechanism.
- This paper gives a detailed mathematical description of their method, showing how to extract local features to measure short-term changes and to model the global correlations to measure the long-term trend using a convolution structure.

Cons:
- Complexity analysis is needed. One of the motivations of this paper is the high complexity of self-attention and this paper tries to reduce the complexity. However, the full text doesn't analyze the complexity of the proposed method, just saying the complexity of the method is O(L), but 'L' is not mentioned in the full text. And the complexity of the method seems to be N*logI.
- Isometric convolution is not explained in detail as the core part of the article. How is isometric reflected? Why do convolutions in this form?
- The paper proves that the proposed Multi-scale Isometric Convolution is superior to self-attention through experiments. However, for the short length of 92, the isometric convolution performance is all lower than self-attention in multivariate forecasting, which is inconsistent with the description of the paper, and there are no further explanations and analysis.

Questions：
- In Section 3.2, what conditions are satisfied with the kernels to ensure that the time series length is constant? How is each kernel determined?
- In Section 3.3, is 'mean' here an average operation? How to average a time series? How to get the result of o-dimension from i-dimension time series through mean operation?
- In section B.3, why is the replacement of Isometric convolution and self-attention in MICN-regre instead of the Seasonal Prediction Block?

**Summary Of The Paper:**

This paper proposes a method named MICN to replace the self-attention module and achieves linear computational complexity. This method is based on convolution structure and proposes Multi-scale Isometric Convolution using multiple branches of different convolution kernels to model different potential pattern information of time series. The MICN has a local-global structure where the local features of the sequence are extracted by a local module based on downsampling convolution and the global correlation is modeled by a global module based on isometric convolution. The results show that MICN achieves state-of-the-art accuracy and isometric convolution is superior in both complexity and effectiveness to self-attention.

**Summary Of The Review:**

The idea is interesting and somewhat novel. More theoretical analysis would be better.

---

> ### Author Response · Authors · 2022-11-12
> **Response to Reviewer PRkd**
>
> We thank the reviewer for their time and feedback on our paper. We respond to some of the reviewer's comments:
>
> ---
> #### **Q1: Complexity analysis.**
> #### **A1:** The complexity of down-sampling convolution is: $ \mathcal{O}(N(i\ast \frac{I+O}{i}\ast D\ast D))=\mathcal{O}(N(I+O)D^{2}) $, where $N$ indicates the number of branches. $I$ is the input sequence length, and $O$ is the prediction length, and $ i \in ( \frac{I}{6}, \frac{I}{8} )$ is the kernel of convolution. $D$ is the hidden state. If we set $ O = mI $, the complexity of down-sampling is $ \mathcal{O}(N(m+1)ID^{2}) $. The complexity of isometric convolution is: $ \mathcal{O}(N(\frac{I}{i})^{2}\ast D\ast D) = \mathcal{O}(Nc^{2}D^{2}) $, where $ c\in( 6,8 ) = \frac{I}{i} $ is a constant. So the complexity of MICN is $ \mathcal{O}(max(N(m+1)ID^{2}, Nc^{2}D^{2} )) = \mathcal{O}(N(m+1)ID^{2}) $. If we set $ L = (I+O) $, the complexity of MICN is $ \mathcal{O}(NLD^{2}) $ , which is linear about the length of sequence to be processed.
> ---
>
> #### **Q2: The explanation of isometric convolution in MICN.**
> #### **A2:** We use the Isometric Convolution for the following reasons: **Firstly**, the current generative prediction approach is to add placeholder to the input sequence (i.e., $concat(X,X_{0})$), which has no actual sequence information in the second half. And the Isometric Convolution is a variant of causal convolution, which can enable sequential inference of sequences and introduces a global temporal pattern bias. **Secondly**, the Isometric convolution is trained from a large amount of data to determine the large convolutional kernel to aggregate the temporal correlations embodied in all training data. It enables parameter sharing across the entire data set and has a strong generalization ability.
> ---
>
> #### **Q3: More analysis about the Isometric Convolution and self-attention.**
> #### **A3:** We conduct more experiments to compare the Isometric Convolution and self-attention on full benchmarks. We have updated our manuscript and shown the results in Appendix B.9. We can conclude that the Isometric Convolution outperforms self-attention in the most cases. And we also note that in some cases self-attention is slightly more effective. We believe that this is related to the corresponding datasets and that we will analyse the datasets in detail in the future.
> ---
>
> #### **Q4: In Section 3.2, what conditions are satisfied with the kernels to ensure that the time series length is constant? How is each kernel determined?**
> #### **A4:** As a rule, we take the values of kernels from $ ( \frac{I}{4}, \frac{I}{6}, \frac{I}{8}... ) $ and set $ stride=kernel $. So after the down-sampling convolution, the isometric convolution and the up-sampling convolution, the length of the output sequence will remain the same. More generally, if the kernel is not divisible by the length of the sequence, we will pad the input sequece and truncate the output sequence to keep the length constant. We have conducted more experiments for this situation and shown the results in Table 20 in Appendix B.9. We can conclude that the different kernels with padding and truncation operation in the local-global module have a relatively small impact on the final results, which indicates that our advanced structure instead of others plays a major role in the performance. And the each kernel is  determined empirically.
> ---
>
> #### **Q5: In Section 3.3, is 'mean' here an average operation? How to average a time series? How to get the result of o-dimension from i-dimension time series through mean operation?**
> #### **A5:** Yes, 'mean' is an average opetation. We mean the time series along the dimension of time, i.e.,($ mean(X_{t}\in R^{I\times d}) = X_{mean}\in R^{1\times d} $ , where $d$ is the dimension of time series. In multivariate time series forecasting, the dimension of the input series is equal to that of the series to be predicted.  In our paper, mean prediction is  $ repeat(mean(X_{t}\in R^{I\times d})) = repeat(X_{mean}\in R^{1\times d}) = Y_{t}^{mean}\in R^{O\times d} $.
> ---
>
> #### **Q6: In section B.3, why is the replacement of Isometric convolution and self-attention in MICN-regre instead of the Seasonal Prediction Block?**
> #### **A6:** The replacement of Isometric convolution and self-attention is in the Seasonal Prediction Block of the model MICN-regre. The Seasonal Prediction Block is a block in our model MICN-regre. And the difference between MICN-regre and MICN-mean lies in the Trend-cyclical Block.
> ---
>
> We hope that our responses have solved your problems. Thank you for your thoughtful comments again.

---

> ### Author Response · Authors · 2022-12-10
> **Further Discussions**
>
> Dear Reviewer PRkd,
>
> Are the answers helpful to your questions? If not, we expect more discussions. Thanks.
>
> All Authors

---

### Official Review · Reviewer_FdbB · 2022-10-23

**Confidence:** 4
**Clarity, Quality, Novelty And Reproducibility:** The paper is well written overall and…
**Correctness:** 4
**Technical Novelty And Significance:** 4
**Empirical Novelty And Significance:** 3
**Recommendation:** 8

**Strength And Weaknesses:**

The topic of this paper address the problem of long-term series forecasting. The topic is very interesting and the paper is well written. The authors provide a comprehensive review of the related papers. Mathematical models and figures are proposed to understand the problem and the solution provided by the authors. Extensive experiments are provided to demonstrate the effectiveness of the proposed method.

**Summary Of The Paper:**

This paper focuses on long term forecasting. The problem to be solved is to predict values of a variable for a future period. The authors start by providing a comprehensive review of the related papers and end with the conclusion that a good forecasting method should have the  ability to extract local features to measure short-term changes, and the ability to model the global
correlations to measure the long-term trend. Based on that, they propose a Multi-scale Isometric Convolution Network (MICN). They use multiple branches of different convolution kernels to model different potential pattern information of the sequence separately.

**Summary Of The Review:**

This paper proposes a Multi-scale Isometric Convolution Network (MICN)  based on convolution structure to efficiently replace the self-attention,and it achieves linear computational complexity and memory cost.
The  empirical studies show that the proposed model improves the performance of state-ofthe-art methods by 18.2% and 24.5% for multivariate and univariate forecasting, respectively

---

> ### Author Response · Authors · 2022-11-18
> **Thanks**
>
> Thanks for your time and positive comments!

---

### Official Review · Reviewer_eyrH · 2022-10-25

**Confidence:** 4
**Correctness:** 3
**Technical Novelty And Significance:** 3
**Empirical Novelty And Significance:** 3
**Recommendation:** 6

**Clarity, Quality, Novelty And Reproducibility:**

The paper is well written with detailed appendix. The general idea of using a hierarchy alike
structure for capturing local-global interaction, plus solving the self-attention complexity issue, are
not completely novel (many works in vision transformer field), but the authors manage to design a
structure specifically for time series forecasting.

**Strength And Weaknesses:**

Strength:
The overall paper is well written and most of the concepts are concisely explained. As the paper
stated, both local patterns and global dependency are crucial for long-term time series
forecasting. The design is well motivated. The benchmark results are also promising to
demonstrate the effectiveness of the framework.

Weaknesses:
Despite the state-of-the-art accuracy, it seems that the main improvement comes from the
regression-based trend-cyclical prediction. Trend-cyclical analysis in section 4.2 is not enough to
understand how the regression influence the prediction. I suggest evaluating the regression and
the mean prediction separately from the full benchmark performance. Also, the results show that
MICN-mean has comparable performance (sometimes even worse) than FEDformer, while
FEDformer is using mean trend prediction. I can’t help but thinking if FEDformer with regression
trend prediction can surpass MICN. More discussion in trend-cyclical prediction is needed.
Another thing that can be discussed more is the intuition behind isometric convolution. The
motivation of the linear complexity is good, but the paper is unclear about how isometric
convolution is going to capture the correlations between local time region. Besides, there exists
many works that focus on the efficiency of self-attention and many has linear complexity as well
(e.g., Linformer [1], Fastformer [2], …). If the primary motivation behind isometric convolution is
the complexity then these works should be discussed as well.
I also have concerns over the explainability of the local-global structure. Self-attention is good at
making interpretable prediction (can be crucial for real-life application, as the paper mentioned),
whereas I don’t see how MICN can be interpreted at a first glance. It is sad when the paper
argues “the framework can deeply mine the intricate temporal patterns” but fails to interpret them.
This is also one of the reasons I am curious about how linear self-attention can perform instead of
isometric convolution.
Other minor suggestion:
1. The abstract says that the structure can model patterns separately and “purposefully”. The
wording is vague and confusing. Does it indicate that we can incorporate external knowledge
into the model?
2. In section 3.1, briefly mentions that the term (e.g., “AvgPool”, “kernel”) is from the convolution
perspective. I first thought the “kernel” represents the kernel function.
3. The difference between ablation table 5 and table 13 can be explained more.
[1] Wang, Sinong, et al. “Linformer: Self-Attention with Linear Complexity.” ArXiv.org, 14 June
2020, https://arxiv.org/abs/2006.04768.
[2] Wu, Chuhan, et al. “Fastformer: Additive Attention Can Be All You Need.” ArXiv.org, 5 Sept.
2021, https://arxiv.org/abs/2108.09084.

**Summary Of The Paper:**

The paper proposes the Multi-scale Isometric Convolution Network, which can efficiently capture
local-global interaction of time series data for long-term forecasting. It uses multi-scaled
convolutions for local information extraction and isometric convolution to capture the global
relationship. In general, it achieves the state-of-the-art accuracy in five time series benchmarks
with linear complexity.

**Summary Of The Review:**

As I mentioned above, the overall method is well motivated and the benchmark results are
promising, whereas the paper can be further improved by more ablation studies and discussions.

---

> ### Author Response · Authors · 2022-11-12
> **Response to Reviewer eyrH**
>
> We thank the reviewer for the constructive feedback and provide responses below.
>
> ---
> #### **Q1: More discussion in trend-cyclical prediction.**
> #### **A1:** We have conducted more experiments about the Trend-cyclical analysis and the results prove the validity of our proposed model. We have updated out paper draft and shown the results in Appendix B.6.
> ---
> #### **Q2: The explanation of isometric convolution in MICN.**
> #### **A2:** We use the Isometric Convolution for the following reasons: **Firstly**, the current generative prediction approach is to add placeholder to the input sequence (i.e., $concat(X,X_{0})$), which has no actual sequence information in the second half. And the Isometric Convolution is a variant of causal convolution, which can enable sequential inference of sequences and introduces a global temporal pattern bias. **Secondly**, the Isometric convolution is trained from a large amount of data to determine the large convolutional kernel to aggregate the temporal correlations embodied in all training data. It enables parameter sharing across the entire data set and has a strong generalization ability.
> ---
> #### **Q3: The explanation of the local-global structure in MICN.**
> #### **A3:** We have updated our manuscript and shown the detailed description of the local-global structure in Appendix B.7.
> ---
> #### **Q4:More discussion about the linear complexity model: Linformer, Fastformer.**
> #### **A4:** In accordance with the reviewer's suggestion, we have conducted more experiments to compare isometric convolution with self-attention, Linformer, Fastformer. The results show that our proposed method performs best. We have updated our manuscript and shown the results in Appendix B.8.
> ---
> #### **Q5: About other minor suggestion.**
> #### **A5:** We thank the reviewer for useful suggestions.
> *1.  The abstract says that the structure can model patterns separately and “purposefully”. The wording is vague and confusing. Does it indicate that we can incorporate external knowledge into the model?*
>
> We will try to take the external knowledge into consideration to complete an interpretable models in the future.
>
> *2.  In section 3.1, briefly mentions that the term (e.g., “AvgPool”, “kernel”) is from the convolution perspective. I first thought the “kernel” represents the kernel function.*
>
> We have carefully revised our manuscript to make it clearer.
>
> ---
> Thank you for your thoughtful comments and suggestions again.

---

> ### Author Response · Authors · 2022-12-10
> **Further Discussions**
>
> Dear Reviewer eyrH,
>
> Are the answers helpful to your questions? If not, we expect more discussions. Thanks.
>
> All Authors

---

### Decision · Program_Chairs · 2023-01-20

**Decision:**

Accept: notable-top-5%

**Justification For Why Not Higher Score:**

The manuscript should be revised to include complexity analysis.

**Justification For Why Not Lower Score:**

The paper presents a transformer based method to capture long term temporal dependencies, while also preserving short-term sequences for long term time-series predictions. It has potential for interpretable time-series forecasting. The paper is clearly written.

**Metareview: Summary, Strengths And Weaknesses:**

The paper presents a transformer model based method for efficient capture of local-global interaction in time series data for long-term forecasting. It proposes a multi-scaled convolutions for extracting short term dependencies and isometric convolution for long term dependencies. The paper has a clear motivation, the algorithm is clearly explained and the experiments are sound. In general, the paper is well-written and is clearly explained.

Authors bold claims on the interpretation have not been sufficiently substantiated. Although authors have indicated that this is a scope of future work, recommend to include these texts subtly without emphasis.

The paper should be revised to include complexity analysis of the proposed method.

**Note From Pc:**

if the above contains the word "oral" or "spotlight" please see: "oral" presentation means -> notable-top-5% and "spotlight" means -> notable-top-25%. As stated in our emails, we are disassociating presentation type from AC recommendations

**Summary Of Ac-Reviewer Meeting:**

NIL